# TrajFlow: Nationwide Pseudo GPS Trajectory Generation with Flow Matching Models

**Peiran Li**[1*]  **Jiawei Wang**[2†]  **Haoran Zhang**[3]  **Xiaodan Shi**[4]

**Noboru Koshizuka**[2]  **Chihiro Shimizu**[1]  **Renhe Jiang**[2]

[1]Hitotsubashi University  [2]The University of Tokyo  [3]LocationMind Inc.
[4]Stockholm University

## ABSTRACT

The importance of mobile phone GPS trajectory data is widely recognized across many fields, yet the use of real data is often hindered by privacy concerns, limited accessibility, and high acquisition costs. As a result, generating pseudo–GPS trajectory data has become an active area of research. Recent diffusion-based approaches have achieved strong fidelity but remain limited in spatial scale (small urban areas), transportation-mode diversity, and efficiency (requiring numerous sampling steps). To address these challenges, we introduce *TrajFlow*, which to the best of our knowledge is the first flow-matching-based generative model for GPS trajectory generation. TrajFlow leverages the flow-matching paradigm to improve robustness and efficiency across multiple geospatial scales, and incorporates a trajectory harmonization & reconstruction strategy to jointly address scalability, diversity, and efficiency. Using a nationwide mobile phone GPS dataset with millions of trajectories across Japan, we show that *TrajFlow* or its variants consistently outperform diffusion-based and deep generative baselines at urban, metropolitan, and nationwide levels. As the first nationwide, multi-scale GPS trajectory generation model, *TrajFlow* demonstrates strong potential to support inter-region urban planning, traffic management, and disaster response, thereby advancing the resilience and intelligence of future mobility systems.

## 1 INTRODUCTION

Human mobility data—particularly the rapidly growing mobile phone GPS data—has been widely applied across diverse domains, including urban studies (Chen et al., 2023; Jin et al., 2023), epidemiological prediction and control (Bao et al., 2020; Zhang et al., 2022), and transportation and travel planning (Torre-Bastida et al., 2018). However, the use of real personal mobility data poses several challenges, such as privacy concerns, limited accessibility, and substantial financial or time costs. Privacy is especially critical, as data collection and utilization may reveal sensitive personal details, which in turn makes human mobility datasets difficult to access and share.

Consequently, recent years have witnessed a growing number of studies on GPS trajectory generation (Zhu et al., 2023b; Jiang et al., 2023; Wei et al., 2024; Zhu et al., 2023a). Notably, several research gaps remain even in state-of-the-art (SOTA) models for this task: **Multi-scale capability:** Existing models primarily focus on urban-level data and struggle to generalize to regional or nationwide scales. This limitation significantly constrains the practical applicability of generated trajectories, as real-world deployment often requires modeling across multiple spatial levels. In particular, maintaining a stable signal-to-noise ratio (SNR) becomes increasingly challenging when extending trajectory generation from block-level movements to citywide or nationwide patterns, underscoring the need for models with robust multi-scale capability. **Transportation-mode diversity:** Current approaches are largely confined to taxi trajectory data. While taxi GPS traces are valuable,

---

*Code open-sourced at `https://github.com/ZeroCSIS/TrajFlow`

†Corresponding author. Email: `jiawei@g.ecc.u-tokyo.ac.jp`

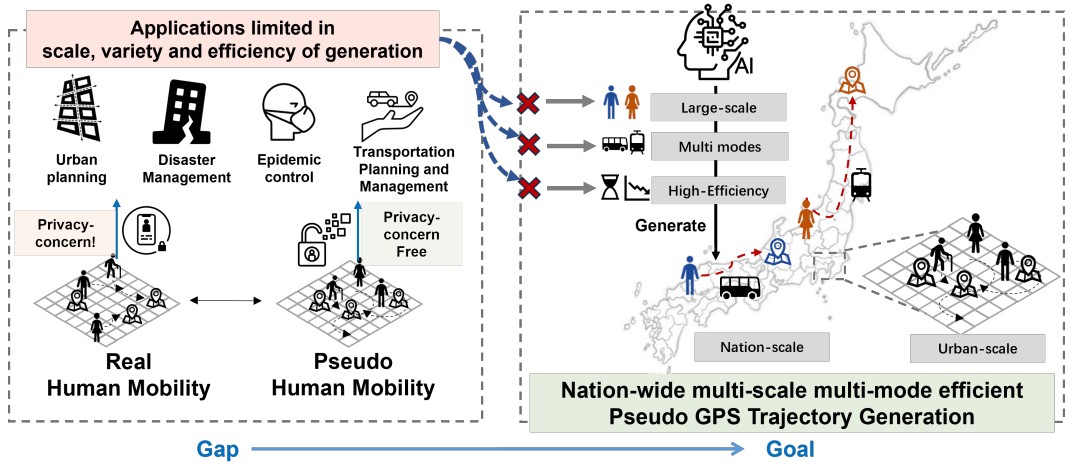

Figure 1: In pseudo-human mobility generation, three key challenges remain to be addressed: multi-scale capability, transportation-mode diversity, and training & inference efficiency.

real human mobility involves a much broader range of transportation modes (e.g., train, car, bike and walk), which are not adequately represented in existing models. **Training/inference efficiency and robustness:** Most SOTA methods rely on diffusion-based frameworks. Although methods such as DDIM (Denoising Diffusion Implicit Models) can accelerate sampling, they still require iterative denoising and remain computationally expensive. Moreover, recent work has shown that commonly used diffusion objectives are closely related to ELBO-based optimization and can be interpreted as weighted integrals of ELBOs over noise levels (Kingma & Gao, 2023). By contrast, Flow Matching directly regresses the target vector field along a prescribed conditional probability path, and, when instantiated with diffusion paths, has been shown to provide a more robust and stable training alternative (Lipman et al., 2022).

To address these limitations, we propose **TrajFlow**, a flow-matching–based GPS trajectory generation model designed to produce nationwide, multi-scale pseudo-GPS trajectories. The main contributions of this study are summarized as follows:

- **Novel paradigm:** We present the first flow-matching–based generative framework for GPS trajectory modeling, and show that the flow-matching paradigm improves robustness and stability across multi-scale scenarios.
- **Methodological design:** We integrate trajectory harmonization, OD-conditioned normalization, and flow-based training into a unified framework that jointly addresses scalability, diversity, and efficiency in GPS trajectory generation.
- **Empirical validation:** Using a nationwide mobile phone GPS dataset with millions of users, we demonstrate that **TrajFlow** achieves state-of-the-art performance across urban, metropolitan, and nationwide settings, highlighting its value for large-scale human mobility modeling.

## 2 RELATED WORK

**Human Mobility Generation.** Human mobility generation has attracted growing attention across both computer science and social science domains (Feng et al., 2020; Simini et al., 2021; Jiawei et al., 2024). In the early stages, before the widespread availability of mobile phone positioning data, research primarily relied on mechanism-based approaches. Travel survey data—often referred to as travel diaries—were commonly used. Such surveys typically record an individual's daily travel activities, providing detailed information including the sequence of mobility activities and the modes of transportation. These rich datasets enabled the development of activity-based models for human mobility sequence generation (Karamshuk et al., 2011; Hess et al., 2015; Barbosa et al., 2018). With the growing demand for large-scale mobility data, researchers began to explore alternative sources such as GPS trajectories and call detail records (CDR). For example, TimeGeo (Jiang et al., 2016) ex-

tended activity-based modeling by leveraging GPS and CDR data. Different from TimeGeo, studies leveraging mobile phone GPS and CDR data have investigated the generation of more realistic activity patterns by combining deep learning models. For instance, Act2Loc (Liu et al., 2024) focuses on activity-to-location generation, while GeoAvatar (Li et al., 2022; 2025) achieves individualized trajectory generation, reflecting both temporal regularity and personal heterogeneity. More recently, advances in large language models (LLMs) have allowed for interpretable mobility generation at a relatively higher computation cost (Jiawei et al., 2024; Feng et al., 2025; Beneduce et al., 2025).

**GPS trajectory generation.**   As one of the most widely used data sources for representing human mobility, GPS trajectories have become a central focus in generative modeling. A variety of approaches have been proposed for GPS trajectory generation, ranging from model-based models to learning-based frameworks (Yin et al., 2017; Hsu et al., 2024; Zhu et al., 2023b; Wang et al., 2021). Early work often adopted sequence modeling techniques such as Hidden Markov Models (HMMs) and Recurrent Neural Networks (RNNs), including their variants like LSTMs and GRUs (Yu et al., 2019). For example, Song et al. (Song et al., 2016) applied an HMM-based model to predict and simulate large-scale human mobility patterns following natural disasters. Beyond Markovian models, deep generative models such as Variational Autoencoders (VAEs) or generative adversarial networks (GANs) have been introduced to capture latent mobility representations (Liu et al., 2018; Huang et al., 2019; Chen et al., 2021). More recently, the rapid proliferation of denoising diffusion probabilistic models (DDPM) have further advanced GPS trajectory generation. DiffTraj (Zhu et al., 2023b) and its extensions (Wei et al., 2024; Zhu et al., 2024) have shown strong performance in generating urban-scale taxi trajectories.

## 3   PRELIMINARY

### 3.1   PROBLEM DEFINITION

**Definition 1** (Human Mobility (GPS Trajectory)). A human mobility record is represented as a tuple $(lat, lon, t)$, indicating that an anonymous user visits a geographic location $l$ (specified by latitude and longitude) at time $t$. A mobility trajectory is then defined as an ordered sequence of such records, $\text{traj} = \{(lat_0, lon_0, t_0), (lat_1, lon_1, t_1), \ldots, (lat_n, lon_n, t_n)\}$. In our setting, anonymized data are used, and user identifiers $u$ are replaced with random tokens to ensure privacy.

**Definition 2** (GPS Trajectory Generation). Given a ground-truth dataset of GPS trajectories $X = \{\text{traj}_x^1, \text{traj}_x^2, \ldots, \text{traj}_x^m\}$, the goal of GPS trajectory generation is to synthesize a pseudo-dataset $Y = \{\text{traj}_y^1, \text{traj}_y^2, \ldots, \text{traj}_y^m\}$, such that $Y$ closely matches the distributional characteristics of $X$ while preserving user privacy.

### 3.2   FLOW MATCHING MODEL

Flow Matching is a powerful class of generative models designed to learn continuous transformations between probability distributions. Unlike diffusion models that rely on a fixed noising process, flow matching models learn a vector field $v_t(x)$ that directly models the "flow" of particles from a simple base distribution $p_0$ (e.g., a standard Gaussian) to a complex target data distribution $p_1$ (e.g., the distribution of real GPS trajectories).

The core idea is to define a probability path $p_t(x)$ and a corresponding time-dependent vector field $v_t(x)$ such that a sample $x_0 \sim p_0$ can be transformed into a sample $x_1 \sim p_1$ by solving the ordinary differential equation (ODE)(Lipman et al., 2022), $\frac{dx_t}{dt} = v_t(x_t)$, where $x_t$ is the state of a particle at time $t \in [0, 1]$. The goal is to train a neural network $v_\theta(x, t)$ to approximate this ground-truth vector field $v_t(x)$.

A key innovation in flow matching is the objective function. Instead of directly regressing on the often-intractable vector field of the marginal probability path $p_t$, Conditional Flow Matching (CFM) defines a conditional probability path $p_t(x \mid x_1)$ and a corresponding conditional vector field $u_t(x \mid x_1)$ that are much simpler to compute. A common choice for the conditional path is a simple linear interpolation between a noise sample $x_0$ and a data sample $x_1$: $p_t(x \mid x_1) = \mathcal{N}(x \mid (1 - t)x_0 + tx_1, \ \sigma^2 I)$. However, the simplest and most common formulation, which we adopt, regresses the model directly on the vector field defined by a straight path between $x_0$ and $x_1$: $u_t(x \mid x_1) = x_1 - x_0$.

The model $v_\theta(x, t)$ is then trained to predict this vector field $u_t$ given the point $x_t$ on the path. The flow matching objective is a simple regression loss:

$$\mathcal{L}_{\text{FM}} = \mathbb{E}_{t, p(x_1), p(x_0)} \left[ \left\| v_\theta\big((1-t)x_0 + tx_1, t\big) - (x_1 - x_0) \right\|^2 \right], \tag{1}$$

where $t$ is sampled uniformly from $[0, 1]$, $x_1$ is sampled from the real data distribution, and $x_0$ is sampled from the prior noise distribution. This objective allows the model to learn the complex data distribution stably and efficiently. Once trained, we can generate new data by sampling a point $x_0 \sim p_0$ and solving the learned ODE $\frac{dx}{dt} = v_\theta(x, t)$ from $t = 0$ to $t = 1$ using an ODE solver.

# 4 TRAJFLOW

## 4.1 MOTIVATION

Diffusion-based models (e.g., DiffTraj) have achieved high-fidelity GPS trajectory generation at the *urban* scale, but when the spatial scale grows, e.g., from urban scale, metropolis scale, and to nationwide scale, we observe a sharp degradation of accuracy across multiple metrics (see Fig. 2a).

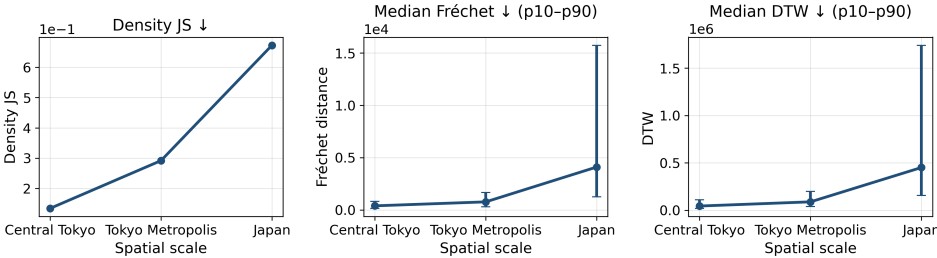

(a) The performance of the DiffTraj degrades during multi-scale generation. Each subplot shows one metric across spatial scales (e.g., Central Tokyo → Tokyo Metropolis → Japan).

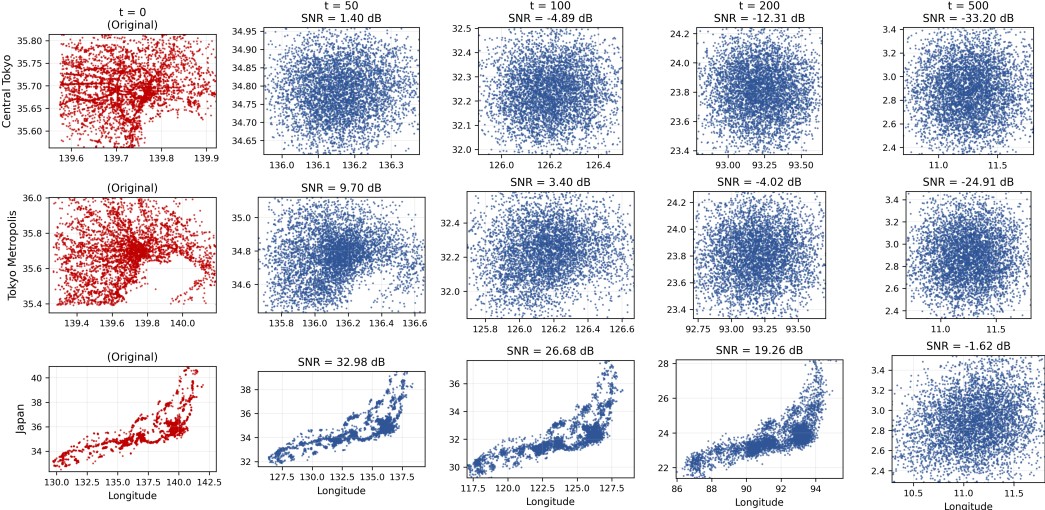

(b) SNR collapses during the diffusion noising procedure. A noise parameter suitable for a nationwide scale is not suitable for smaller regions (e.g., Tokyo Metropolis).

Figure 2: **(a)** shows the accuracy degradation of DiffTraj at increasing spatial scales. **(b)** shows the SNR collapses when applying a fixed noising parameter across scales.

We suggest that there are two key reasons why diffusion-based SOTA models fail to generalize across scales. First, when expanding to larger geographical regions, the subsets of fine-grained local trajectories become small in magnitude, resulting in an extremely low SNR. In these cases, the weak signal is easily overwhelmed, forcing the reverse process to reconstruct fine-grained structures from

highly noisy inputs. Standard mean squared error (MSE) objectives further intensify this challenge by prioritizing absolute rather than relative error, thereby aggravating the imbalance across scales. To address this, *data harmonization becomes essential*: by curating and rescaling data distributions across different spatial scales, we can explicitly compensate for this imbalance and provide a more balanced learning signal (see Sec. 4.2). Second, diffusion models rely on a stepwise denoising process that gradually transforms Gaussian noise into samples. In conditional settings where subsets of data span vastly different numerical ranges—for example, micro-scale local trips versus macro-scale long-distance journeys—the forward noising process injects noise of nearly fixed magnitude regardless of scale. This scale-invariant noise injection leads to a fundamental mismatch that reduces model robustness. To overcome this limitation, we adopt *flow matching* as a more suitable alternative: by directly learning the continuous probability flow between a simple prior and the target distribution, it sidesteps the fixed-step denoising chain and provides a more flexible generative mechanism for multi-scale settings (see Secs. 4.3 and 4.4). In the following, we detail how our proposed framework, TrajFlow, addresses the critical challenges of urban mobility modeling and generates reliable pseudo-GPS trajectories.

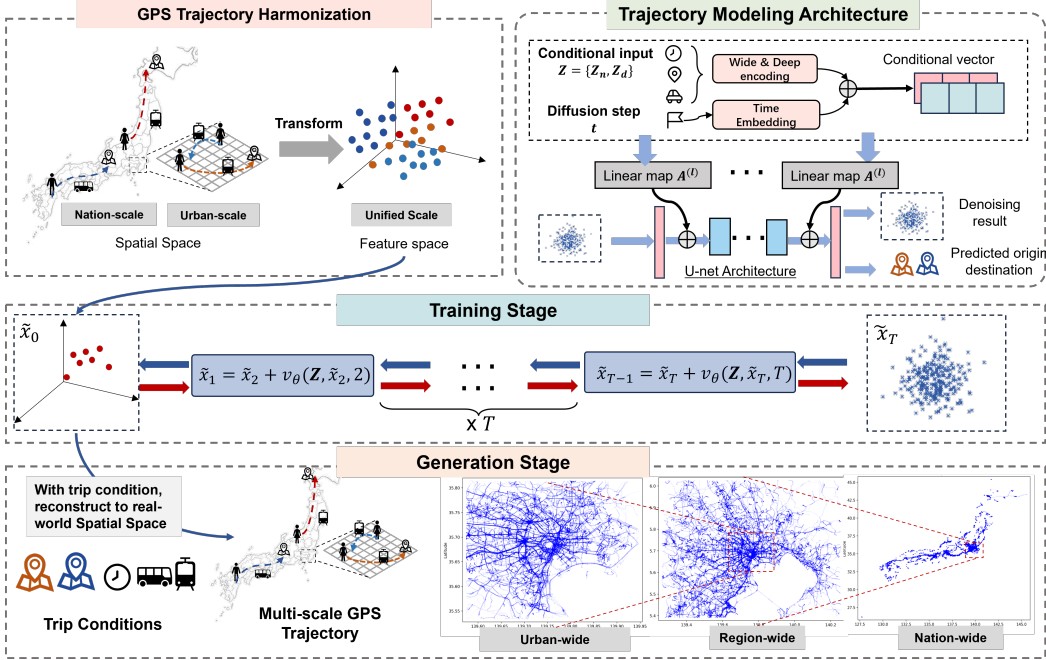

Figure 3: The overview of the proposed TrajFlow.

## 4.2 MULTI-SCALE GPS TRAJECTORY HARMONIZATION AND RECONSTRUCTION

To address the SNR imbalance across scales, we adopt a trajectory harmonization & reconstruction strategy in conjunction with the flow matching framework (see the upper-left panel of Fig. 3). Rather than working directly on raw GPS coordinates, which vary across orders of magnitude, we normalize each trajectory individually, rescaling all points into a shared bounded coordinate space. The model thus operates in this normalized space, predicts fine OD location within the given region-level OD zones, together with waypoints under flow matching; afterward, we invert the normalization back to real geographic coordinates. This prevents tiny local displacements from being overwhelmed by large-scale variations, stabilizes gradient magnitudes during optimization, and accelerates convergence (Ioffe & Szegedy, 2015).

In addition, to enhance the efficiency and stability of model learning, we apply a *trajectory-feature transformation* step that simplifies each trajectory while preserving its essential geometric structure. This kind of trajectory-feature transformation reduces the length of the raw trajectories from $L$ to $D$, where $D$ is substantially smaller than $L$, thereby lowering computational overhead and improving training stability. In practice, the process involves recursively removing points that fall

within a tolerance $\epsilon$ of the line segment formed by their neighbors, retaining only those points that contribute to the overall geometric shape. As a result, long straight segments are compressed into a few representative points, while turning points or areas of high curvature are preserved. In this study, we experiment with multiple harmonization methods and find that the classical *Ramer–Douglas–Peucker* (RDP) algorithm (Ramer, 1972) provides the best trade-off between compression and fidelity. Details of this algorithm are presented in the Appendix. A.2.

The proposed strategy can be interpreted as analogous to the normalization–denormalization procedure in feature preprocessing: trajectories are first compressed into a compact representation to facilitate efficient model training, and subsequently expanded back to their original scale for deployment, see Appendix A.2.

## 4.3 GPS TRAJECTORY MODELING ARCHITECTURE

As shown in the upper-right panel of Fig. 3, we adopt the Wide & Deep module (Zhu et al., 2023b) to embed the input condition of a trajectory, including departure time, OD (zone-level), and transportation mode. Concretely, the conditional vector $e_c$ is formed by fusing a linear wide projection of numeric motion/context characteristics with a non-linear deep projection of discrete context characteristics, and the result $e_c$ is injected into every block of the vector field network $v_\theta(x_t, t, e_c)$ that parameterizes our probability flow:

$$\frac{dx_t}{dt} = v_\theta(x_t, t, e_c). \tag{2}$$

*Inputs and feature partition.* The first part of the input is numeric features $\boldsymbol{Z}_n$: trajectory-level scalars such as average speed, average inter-point distance, elapsed time, cumulative distance/steps (all standardized). The other part is the discrete features $\boldsymbol{Z}_d$, including departure time of a day, transportation mode, and OD (zone-level).

*Wide&Deep encoding.* We use a linear projection to map the numeric features $\boldsymbol{Z}_n$ into an $n$-dimensional vector, denoted as $e_{\text{wide}}$. Each discrete feature in $\boldsymbol{Z}_d$ is first embedded into an $n$-dimensional vector; the resulting embeddings are concatenated and passed through two MLP layers with nonlinearity to produce $e_{\text{deep}}$. The final condition vector is obtained by fusing the two branches:

$$e_c = \text{LayerNorm}(e_{\text{wide}} + e_{\text{deep}}). \tag{3}$$

*Time embedding.* The continuous flow time $t \in [0, 1]$ is encoded by a sinusoidal/Fourier mapping followed by a small MLP to yield a time vector $e_t \in \mathbb{R}^{d_e}$. We combine it with the condition as a single control signal $\tilde{e} = e_c + e_t$.

*Conditional control into the backbone.* Let $h^{(\ell)}$ denote the hidden state of block $\ell$. We broadcast $\tilde{e}$ to all blocks and inject it as an additive, learnable bias as:

$$h^{(\ell)} \leftarrow f^{(\ell)}\big(h^{(\ell)}\big) + A^{(\ell)}\tilde{e} \qquad (\ell = 1, \dots, L), \tag{4}$$

where $f^{(\ell)}$ is the block's transformation and $A^{(\ell)}$ is a learned linear map. This keeps the conditioning pathway lightweight and consistent with the ODE parameterization of the vector field.

## 4.4 FLOW-MATCHING TRAINING AND INFERENCE

At the training stage (i.e., the middle panel of Fig. 3), we use mini-batches of ground-truth trajectories that are first harmonized with RDP and then padded to a common length using a validity mask. For each trajectory, we compute the numeric and discrete conditions, obtain the condition embedding $e_c$ through the Wide&Deep module (Eq. 3), and sample a flow time $t$ with a noise endpoint $x_0$. The straight-path point $x_t$ and the corresponding target conditional vector field are then constructed according to the flow-matching objective (Eq. 1). In addition, we introduce an auxiliary supervised loss between the predicted and ground-truth fine OD location to enhance spatial and semantic awareness. Model parameters are optimized with a masked regression loss over valid tokens, optionally augmented by smoothness and bounded-support regularizers, followed by standard optimizer updates.

At the inference stage (i.e., the last panel of Fig. 3), given a trajectory condition (e.g., departure time, OD (zone-level), transportation mode), we first compute $e_c$ using the Wide&Deep encoder and

prepare the time-embedding schedule. We then initialize $x_0$ from the base distribution (e.g., standard Gaussian noise). The synthetic trajectory is obtained by numerically integrating the learned flow ODE on $[0, 1]$ (Eq. 2), injecting $e_c$ at each solver step as in Eq. 4. The final state $x_1$ is subsequently post-processed (e.g., uniform resampling, mapping back to geographic coordinates, and trimming or interpolating to satisfy length priors), yielding the generated trajectory under the specified condition.

## 5 EXPERIMENTS

In this section, we evaluate **TrajFlow** to answer the following research questions: **Q1:** Does TrajFlow outperform baseline methods on key evaluation metrics? **Q2:** How well does TrajFlow perform in multi-scale trajectory generation? **Q3:** To what extent can TrajFlow reproduce transportation-mode diversity? **Q4:** How efficient is TrajFlow in terms of training and inference?

### 5.1 SETTINGS

**Dataset.** We use the 2023 nationwide Blogwatcher dataset for Japan. This private dataset, provided by Blogwatcher Inc., contains fine-grained GPS records collected through multiple mobile applications and comprises millions of trajectories. Each record includes anonymized user IDs, latitude, longitude, timestamps, and transportation modes.

**Baselines.** Diffusion-based methods represent the current state of the art in GPS trajectory generation, and we adopt *DiffTraj* (Zhu et al., 2023b) as the representative diffusion baseline. In addition, we include *TrajVAE* (Chen et al., 2021) and *TrajGAN* (Rao et al., 2020) as representative deep generative and adversarial baselines, respectively. To disentangle the contributions of the key components in *TrajFlow*, we further evaluate several ablated variants. *w/o-FM:* the CFM objective and ODE solver are replaced with a standard DDPM denoising procedure. Because the performance of a DDPM-style model is highly sensitive to the number of denoising steps, this comparison is reported separately under different step settings in Appendix B. *w/o-OD:* the fine-grained OD prediction head, which predicts origin/destination coordinates within the conditioned OD zones, and the per-trajectory harmonization/reconstruction pipeline are removed, so trajectories are generated directly in geographic space. *w/o-RDP:* the RDP trajectory-harmonization step is omitted, and raw trajectories are fed into the model. We also evaluate the combined removal of RDP and OD prediction. These ablations isolate the effects of the flow-matching objective, the harmonization/reconstruction strategy, and the trajectory-compression step on fidelity and scalability.

**Evaluation Metrics.** We evaluate generation quality from two perspectives. **Aggregated–level:** The *Jensen–Shannon divergence of spatial density (density JS)* measures how well synthetic data reproduces the population-level geo–distribution of movements. **Trajectory–level:** We assess trajectory similarity using *Dynamic Time Warping* (DTW) and the continuous *Fréchet distance* (Fr). DTW captures similarity under flexible alignment between trajectory points, whereas Fréchet distance is more sensitive to the overall geometric shape of the path. Together, they provide complementary evaluations of trajectory fidelity. For both metrics, we report the median, and 10th/90th percentiles (P10/P90) to capture central accuracy and dispersion. Details are provided in Appendix A.1.

### 5.2 EVALUATION

**Overall Performance (Q1).** Across spatial scales - Central Tokyo (urban), Tokyo Metropolis (metro), and Japan (nationwide) - the best-performing configurations consistently come from the *TrajFlow* family, rather than from diffusion or other deep generative baselines. The advantage is more moderate at the urban and metropolitan scales but becomes pronounced nationwide, highlighting stronger cross-scale generalization.

At the urban scale, although the full *TrajFlow* model achieves strong overall performance, the best results are obtained by *TrajFlow*-w/o-RDP&OD, which remains within the flow-matching family. This suggests that when trajectories stay within a relatively homogeneous small-scale spatial range, directly generating raw coordinates may already be sufficient, and the additional RDP-based harmonization/reconstruction is not always necessary.

At the metropolitan scale, several of the best indicators shift from *TrajFlow*-w/o-RDP&OD to *TrajFlow*-w/o-RDP, indicating that the OD-guided harmonization and reconstruction component begins

Table 1: Evaluation grouped by region (unit = km). Best metric in **bold**.

| Method | Density JS ↓ | DTW$_{med}$ ↓ | Fr$_{med}$ ↓ | DTW$_{IQR}$ ↓ | DTW$_{P10}$ ↓ | DTW$_{P90}$ ↓ | Fr$_{IQR}$ ↓ | Fr$_{P10}$ ↓ | Fr$_{P90}$ ↓ |
|---|---|---|---|---|---|---|---|---|---|
| *Central Tokyo* | | | | | | | | | |
| TrajFlow (ours) | 0.0674 | 20.350 | 0.304 | 13.392 | 10.574 | 39.119 | 0.174 | 0.200 | 0.674 |
| TrajFlow-w/o-OD | 0.3560 | 916.436 | 13.862 | 708.183 | 432.567 | 1,740.890 | 7.313 | 7.064 | 20.858 |
| TrajFlow-w/o-RDP | 0.0642 | 22.088 | 0.340 | 16.491 | 11.149 | 47.959 | 0.238 | 0.209 | 0.873 |
| TrajFlow-w/o RDP & OD | **0.0323** | **8.179** | **0.184** | **4.586** | **4.994** | **14.159** | **0.118** | **0.110** | **0.363** |
| DiffTraj | 0.1340 | 44.321 | 0.651 | 40.713 | 19.399 | 109.349 | 0.544 | 0.341 | 1.774 |
| TrajGAN | 0.3087 | 292.430 | 4.442 | 448.839 | 119.230 | 1,288.929 | 7.660 | 1.606 | 21.477 |
| TrajVAE | 0.1041 | 32.874 | 0.469 | 36.387 | 14.000 | 103.232 | 0.679 | 0.258 | 1.842 |
| *Tokyo Metropolis* | | | | | | | | | |
| TrajFlow (ours) | 0.1239 | 18.167 | 0.335 | 16.892 | 7.678 | 44.316 | 0.333 | 0.130 | 0.933 |
| TrajFlow-w/o-OD | 0.1064 | 16.466 | 0.307 | 10.189 | 9.307 | 32.126 | 0.192 | 0.180 | 0.683 |
| TrajFlow-w/o-RDP | 0.1197 | 18.417 | **0.298** | 24.152 | **6.637** | 67.674 | 0.473 | **0.121** | 1.339 |
| TrajFlow-w/o RDP & OD | **0.0800** | **14.416** | 0.303 | **7.684** | 8.659 | **23.978** | **0.188** | 0.176 | **0.592** |
| DiffTraj | 0.2918 | 88.559 | 1.220 | 78.339 | 38.663 | 199.501 | 0.982 | 0.586 | 3.035 |
| TrajGAN | 0.3821 | 604.399 | 10.224 | 1,184.077 | 155.401 | 2,854.060 | 20.627 | 2.290 | 51.389 |
| TrajVAE | 0.1930 | 46.363 | 0.765 | 54.484 | 16.234 | 122.556 | 1.006 | 0.250 | 2.299 |
| *Japan nationwide* | | | | | | | | | |
| TrajFlow (ours) | **0.2270** | **10.977** | **0.192** | **18.221** | **3.984** | **55.964** | **0.361** | **0.072** | **1.119** |
| TrajFlow-w/o-OD | 0.4888 | 100.271 | 1.522 | 75.774 | 50.877 | 216.168 | 1.177 | 0.774 | 3.145 |
| TrajFlow-w/o-RDP | 0.2734 | 24.690 | 0.400 | 27.928 | 9.047 | 92.641 | 0.511 | 0.156 | 1.699 |
| TrajFlow-w/o RDP & OD | 0.4865 | 105.011 | 1.662 | 89.509 | 53.549 | 280.092 | 1.380 | 0.870 | 3.802 |
| DiffTraj | 0.6727 | 451.042 | 5.329 | 635.120 | 157.379 | 1,741.025 | 6.973 | 1.915 | 18.924 |
| TrajGAN | 0.5278 | 403.282 | 6.653 | 999.210 | 79.556 | 2,853.557 | 17.703 | 1.134 | 48.838 |
| TrajVAE | 0.5228 | 135.377 | 2.216 | 236.143 | 28.394 | 577.139 | 3.884 | 0.435 | 9.919 |

to show clear benefits as the spatial range increases. Meanwhile, *TrajFlow* maintains performance comparable to that at the urban scale.

At the nationwide scale, the advantage becomes substantial: DTW$_{med}$ = 10.977, Fr$_{med}$ = 0.192, tight IQR/P90, and the lowest density JS, while alternative methods exhibit much larger medians and heavier tails. Although *TrajFlow* is not strictly the best on every metric at smaller scales, it is the only method family that maintains a strong balance among shape fidelity, stability (IQR/P90), and spatial distribution (density JS) across all scales, and it clearly dominates in the nationwide setting.

Geographic visualizations are provided in Fig. 4. This pattern is consistent with the analysis in Sec. 4.1: when trajectories remain within a relatively small spatial range, the SNR issue in DDPM is less severe; however, under nationwide generation—where urban-, metropolitan-, and nationwide-scale patterns are mixed—the scale mismatch in diffusion hinders performance across scales, a limitation that flow matching addresses more effectively.

**Multi-scale Robustness (Q2).** From the city level to the metropolitan level and further to the nationwide level, *TrajFlow* maintains low median DTW/Fréchet values and relatively narrow dispersion while keeping density JS under control. This result indicates that the model remains robust as the spatial range expands and the heterogeneity of trajectories increases. By contrast, diffusion-based generation becomes increasingly unstable when trajectory subsets span very different numerical ranges, from short local trips to long-distance nationwide movements. This observation is consistent with the analysis in Sec. 4.1: the fixed-magnitude noising process in DDPM introduces a scale mismatch under multi-scale settings, whereas conditional flow matching learns a deterministic transport field and avoids error accumulation along long denoising chains, leading to better robustness across heterogeneous regions and transportation conditions.

**Transportation-mode Diversity (Q3).** Preserving the diversity of transportation modes is as important as maintaining spatial fidelity in generative mobility models.

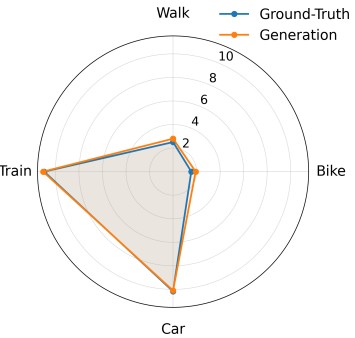

Figure 5: Per-mode average trip distance in Tokyo: ground truth vs. generated.

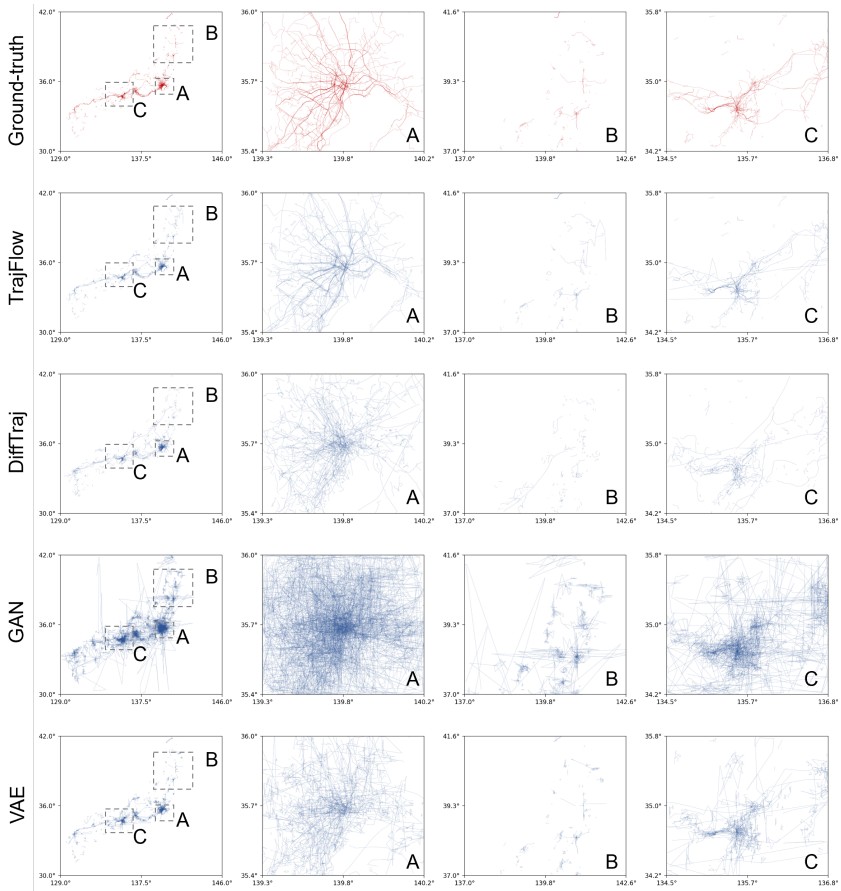

Figure 4: Visualization of trajectory samples. Ground-truth and generated nationwide trajectories are shown with zoomed views highlighting three representative regions: (A) Tokyo Metropolis, (B) Tohoku Area, and (C) Kansai Area, across all generative models.

We evaluate this by comparing per–mode trip distance distributions of ground-truth and generated trajectories in Tokyo. As shown in Fig. 5, the generated data matches the characteristic profiles of four representative modes, maintaining realistic differences in trip lengths. This indicates that *TrajFlow* not only captures spatial fidelity but also preserves transportation-mode diversity (see additional results in Appendix C).

**Training and Inference Efficiency (Q4).** *TrajFlow* is optimized with the CFM objective, which requires only a single time step per sample, and generates trajectories by integrating the learned ODE with a small fixed budget of about 10 steps. In contrast, DDPM performance is highly sensitive to the number of denoising steps (see Appendix B): among the tested settings (10, 50, 100, 200, 300), the best trade-off between Jensen–Shannon divergence and trajectory-shape fidelity typically appears near 200 steps, while 300 steps provide only marginal or inconsistent improvements at significantly higher cost. Notably, even with 300 steps, DDPM fails to reach the performance of *TrajFlow*. These results highlight that competitive fidelity can be achieved by *TrajFlow* with far fewer steps (about 10) in both training and inference.

## 5.3 ABLATION STUDIES

**w/o-FM:** Replacing the CFM objective with a DDPM-style denoising procedure yields monotonic but limited gains as the number of denoising steps increases, and consistently underperforms *TrajFlow* in DTW, Fréchet, and density JS—even with large step budgets. This confirms flow matching as the main driver of fidelity and stability. **w/o-RDP:** Removing trajectory harmonization retains micro-jitter and redundant points, increasing Fréchet/DTW medians and tail metrics; on the Japan

split, it also raises density JS. Thus, RDP improves not only efficiency but also accuracy by suppressing noise and harmonizing sampling. **w/o-OD:** Dropping the OD predictor sometimes lowers density JS in small regions but consistently worsens DTW/Fréchet and dispersion, and degrades both shape and density metrics nationwide. **w/o-RDP & OD:** Combining both removals produces the greatest deterioration, underscoring their important role, particularly on a nationwide scale.

# 6 CONCLUSION

We introduced **TrajFlow**, a flow–matching–based framework for generating pseudo-GPS trajectories. By integrating trajectory harmonization and reconstruction into a conditional flow-matching generative framework, TrajFlow addresses key challenges in large-scale GPS trajectory generation by maintaining robustness across spatial scales, preserving transportation-mode diversity, and achieving strong efficiency, with the clearest advantage at the nationwide scale. Evaluated on a nationwide mobile-phone GPS dataset from Japan, TrajFlow or its variants outperform diffusion and other deep generative baselines. Particularly, it balances trajectory-shape fidelity, stability, and population-level spatial consistency across scales, achieves competitive accuracy with substantially fewer ODE steps.

Due to privacy constraints and data-access limitations, we focus on GPS trajectory generation and do not use any per-user attributes (e.g., age, gender, home–work identifiers) or persistent pseudonymous IDs, and thus the current model does not capture individual preferences. As future work, we aim to extend TrajFlow from GPS trajectory generation to broader human mobility modeling while preserving user privacy.

## LLM USAGE STATEMENT

In accordance with the ICLR 2026 policy on LLM usage, we disclose that LLMs (specifically OpenAI's ChatGPT) were employed as a general-purpose writing assistant. The usage was limited to improving grammar, clarity, and LaTeX formatting of the manuscript.

## ACKNOWLEDGEMENTS

This work was supported by JSPS KAKENHI Grant Number 24K17367, JSPS KAKENHI Grant Number JP25K21264, JSPS KAKENHI JP24K02996, and JST CREST JPMJCR21M2. The authors would like to thank Prof. Ryosuke Shibasaki, of The University of Tokyo and LocationMind Inc., for his support and guidance.

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

# A APPENDIX

## A.1 EVALUATION METRICS

To quantify the fidelity of the generated trajectories, besides the use of density JS-divergence, we also employed two trajectory(line)-level evaluation metrics: DTW and Fréchet distance. DTW aligns trajectories in time and is robust to local temporal shifts, highlighting point-wise spatiotemporal fidelity. Fréchet distance, in contrast, evaluates the closest continuous matching of two curves and is more sensitive to global path geometry. Using both metrics provides a comprehensive view: DTW captures fine-grained temporal accuracy, while Fréchet complements it by emphasizing overall spatial shape consistency. The definitions are shown as follows:

**Aggregated–level: Jensen–Shannon (JS) Divergence of Spatial Density.** Let $p(\mathbf{s})$ and $q(\mathbf{s})$ be the normalized 2-D spatial density maps (kernel–smoothed or mesh–count histograms) of the real and generated trajectories, respectively. The JS divergence is the symmetrized and smoothed version of the Kullback–Leibler divergence:

$$\mathrm{JS}(p \parallel q) = \tfrac{1}{2}\,\mathrm{KL}\big(p \,\big\|\, \tfrac{p+q}{2}\big) + \tfrac{1}{2}\,\mathrm{KL}\big(q \,\big\|\, \tfrac{p+q}{2}\big)\,, \tag{5}$$

where $\mathrm{KL}(p\|m) = \sum_{\mathbf{s}} p(\mathbf{s}) \log \frac{p(\mathbf{s})}{m(\mathbf{s})}$. This metric measures how well the synthetic data reproduces the population-level geographical distribution of movements; lower values indicate closer global density.

**Trajectory–level: Dynamic Time Warping (DTW).** Given two trajectories $A = \{a_1, \ldots, a_m\}$ and $B = \{b_1, \ldots, b_n\}$, where $a_i, b_j \in \mathbb{R}^2$ are latitude–longitude points, DTW finds an alignment path $\pi = \{(i_k, j_k)\}_{k=1}^K$ with $i_1 = j_1 = 1$ and $i_K = m$, $j_K = n$, that minimizes the cumulative Euclidean distance

$$\mathrm{DTW}(A, B) = \min_{\pi} \left\{ \sum_{k=1}^K \|a_{i_k} - b_{j_k}\|_2 \right\}. \tag{6}$$

The alignment allows non-linear warping in the time dimension, making DTW robust to local speed or sampling differences and highlighting fine-grained spatiotemporal fidelity.

**Trajectory–level: Continuous Fréchet Distance.** For the same curves $A$ and $B$ interpreted as continuous functions $\alpha, \beta : [0, 1] \to \mathbb{R}^2$, The Fréchet distance is

$$\mathrm{Fr}(A, B) = \inf_{\phi, \psi}\; \max_{t \in [0,1]} \big\|\alpha\big(\phi(t)\big) - \beta\big(\psi(t)\big)\big\|_2\,, \tag{7}$$

where $\phi$ and $\psi$ range over all continuous, non-decreasing re-parameterizations of $[0, 1]$. Intuitively, it is the minimum leash length required for a person and a dog to walk along the two curves without backtracking. Unlike DTW, Fréchet does not explicitly align discrete time stamps; it captures overall geometric similarity and is sensitive to global path shape.

**Summary of Usage.** For every generated trajectory we compute DTW and Fréchet distances to its nearest ground-truth neighbor and report the **median**, and **10th/90th percentiles (P10/P90)** to capture both typical accuracy and dispersion of errors across the distribution.

Fréchet distances capture geometric deviations, whereas DTW captures temporal misalignment. DTW explicitly aligns sequences in the time dimension - it penalizes trajectories that are spatially similar but temporally mismatched. Computing these metrics enables us to capture both geometry and temporal mismatching.

In addition, we report P10 and P90 alongside the median to capture the model's robustness across the long-tailed complexity of human mobility: human mobility is usually long-tail, reporting only the Median (Central Accuracy) may mask model failures on complex outliers. By reporting P10/P90 (extreme/long-tail case), we demonstrate that TrajFlow remains stable and accurate even for the long-tail part trajectories, rather than just fitting the easy majority.

Together, JS divergence evaluates aggregated-level spatial consistency, while DTW and Fréchet jointly measure individual-path fidelity from complementary temporal and geometric perspectives.

## A.2 Data Compression Algorithms

### A.2.1 Candidate Algorithms

We evaluate seven representative parameterization (data–harmonization) methods implemented in our codebase:

a) **direct_k**: Arc–length reparameterization followed by uniform sampling of $K$ points along $[0, 1]$.

b) **dct**: Discrete Cosine Transform (DCT-II) of the arc–length–uniform curve for $x(t)$ and $y(t)$; keep the first $DCT\_M$ coefficients.

c) **rdp_k**: Ramer–Douglas–Peucker harmonization with a binary search on the tolerance so that the simplified polyline contains approximately $K$ vertices; if necessary, linearly interpolate to exactly $K$ points.

d) **anchor**: Least–squares spline fitting with automatically detected "anchors" (including endpoints) that receive a large weight $w_{\text{anchor}}$ in the fitting.

e) **spline_lsq**: Least–squares spline fitting with uniform weights (no anchors).

f) **dct_deviation**: Take the straight line connecting start and end points as a baseline; apply DCT to the perpendicular deviation sequence and retain $2K - 4$ coefficients together with the endpoints.

g) **fft_complex**: Represent the curve as $z = x + iy$, apply FFT, and keep the first $K$ complex coefficients.

The RDP algorithm simplifies a trajectory by recursively removing points that lie within a distance threshold $\epsilon$ of the line segment connecting their neighboring points, while preserving the essential geometric shape of the trajectory. Formally, given the ground-truth GPS trajectory dataset $\mathcal{X} = \left\{ \text{traj}_x^1, \text{traj}_x^2, \ldots, \text{traj}_x^m \right\}$, each trajectory $\text{traj} = (l_0, l_1, \ldots, l_n)$ is processed as follows:

---

**Algorithm 1** Ramer–Douglas–Peucker (RDP) Algorithm

---

1: **function** RDP($traj, \epsilon$)
2: $\quad d_{max} \leftarrow 0$
3: $\quad index \leftarrow 0$
4: $\quad end \leftarrow \text{length}(traj) - 1$
5: $\quad$ **for** $i = 1$ **to** $end - 1$ **do**
6: $\quad\quad d \leftarrow \text{perpendicularDistance}(traj[i], \text{line}(traj[0], traj[end]))$
7: $\quad\quad$ **if** $d > d_{max}$ **then**
8: $\quad\quad\quad index \leftarrow i$
9: $\quad\quad\quad d_{max} \leftarrow d$
10: $\quad\quad$ **end if**
11: $\quad$ **end for**
12: $\quad$ **if** $d_{max} > \epsilon$ **then**
13: $\quad\quad results1 \leftarrow \text{RDP}(traj[0 \ldots index], \epsilon)$
14: $\quad\quad results2 \leftarrow \text{RDP}(traj[index \ldots end], \epsilon)$
15: $\quad\quad$ **return** concatenate($results1[0 \cdots - 1], results2$)
16: $\quad$ **else**
17: $\quad\quad$ **return** $[traj[0], traj[end]]$
18: $\quad$ **end if**
19: **end function**

---

### A.2.2 Experiments with Different Parameters

To evaluate the influence of trajectory–harmonization granularity, we compared seven parameterization methods—direct_k, dct, rdp_k, anchor, spline_lsq, dct_deviation, and fft_complex—under four target point budgets ($K \in \{5, 10, 20, 30\}$) — while the original length is 120, representing compression ratios of approximately $\{4.2\%, 8.3\%, 16.7\%, 25.0\%\}$ respectively.

For each configuration we report the mean Dynamic Time Warping (DTW) and Fréchet distance (Table 2), and visualize qualitative differences in Figures. 6 & 7 & 8 & 9.

Overall, RDP-based simplification (`rdp_k`) consistently achieves the lowest trajectory-level errors, with average DTW decreasing from $\approx 2.77$ at $K=5$ to $\approx 0.59$ at $K=30$, and Fréchet distance dropping from $\approx 0.058$ to $\approx 0.015$. The closely related spline least–squares (`spline_lsq`) and DCT methods follow as the next best performers. Fourier–based `fft_complex` and deviation–only DCT are markedly worse, reflecting their sensitivity to high–frequency noise.

Increasing the point budget unsurprisingly improves accuracy for all algorithms, but the gains taper beyond $K=20$: DTW and Fréchet for `rdp_k` improve only marginally from $K=20$ to $K=30$. Visual inspection (Figures. 6 & 7 & 8 & 9) confirms that $K=10$, i.e., 8.3% compression rate, already captures the salient geometry of typical trajectories while preserving the strong compression benefits required for efficient training. Based on this analysis, we adopt RDP with $K=10$ as the default harmonization setting for TrajFlow, striking a practical balance between trajectory fidelity and computational cost.

We highlight two necessities of RDP in our tasks:

- RDP is used only as a dimensionality-reduction method. We compress each raw trajectory (often around 120 points) into a smaller set of keypoints (around 10). This compact representation greatly improves flow-matching training efficiency and stability, especially across multiple spatial scales.

- RDP provides the best balance between compression ratio and reconstruction accuracy. As shown in our ablation study, RDP achieves lower DTW and Fréchet distances than alternatives such as DCT, spline fitting, or FFT. This ensures that the simplified representation maintains high spatial fidelity before the interpolation step.

Table 2: Reconstruction Performance (Avg DTW/Fréchet, unit=km). Best (lowest) values in **bold**.

| Method | Param 5 | | Param 10 | | Param 20 | | Param 30 | |
|---|---|---|---|---|---|---|---|---|
| | DTW | Fréchet | DTW | Fréchet | DTW | Fréchet | DTW | Fréchet |
| direct_k | 3.78 | 0.0908 | 1.54 | 0.0430 | 0.79 | 0.0205 | 0.66 | 0.0155 |
| dct | 3.23 | 0.0814 | 1.24 | 0.0326 | 0.69 | 0.0150 | 0.61 | 0.0137 |
| **rdp_k** | **2.77** | **0.0580** | **0.95** | **0.0219** | **0.61** | 0.0143 | **0.59** | 0.0150 |
| anchor | 16.42 | 0.3333 | 1.75 | 0.0405 | 0.76 | 0.0191 | 0.61 | 0.0138 |
| spline_lsq | 3.11 | 0.0665 | 1.22 | 0.0271 | 0.68 | 0.0150 | 0.61 | **0.0136** |
| dct_deviation | 4.66 | 0.0951 | 4.06 | 0.0874 | 4.15 | 0.0842 | 5.17 | 0.1025 |
| fft_complex | 28.32 | 0.4655 | 24.11 | 0.5120 | 25.59 | 0.6054 | 25.43 | 0.6363 |

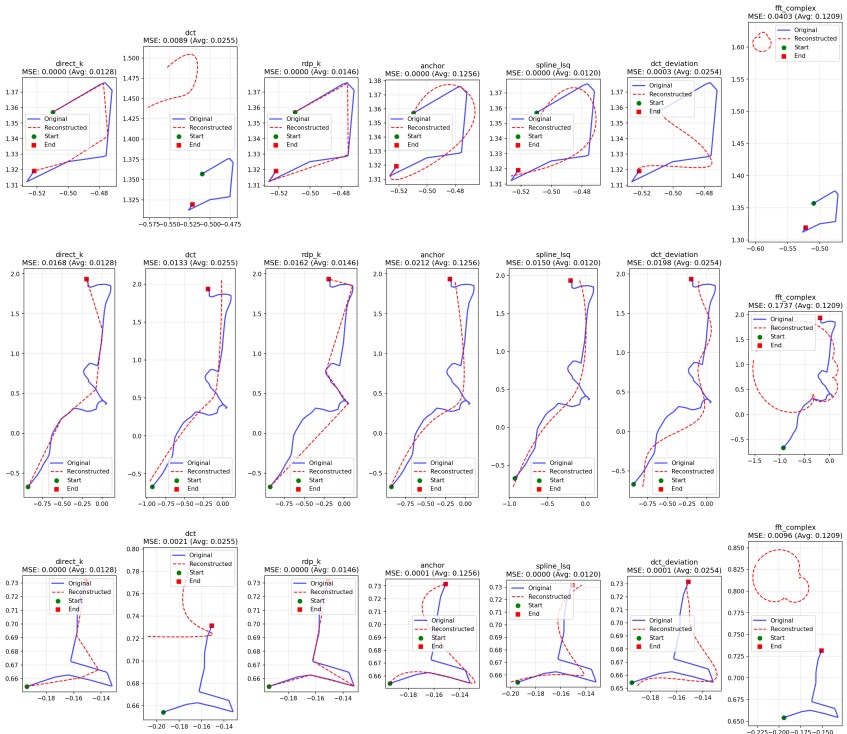

Figure 6: Parameter Number = 5

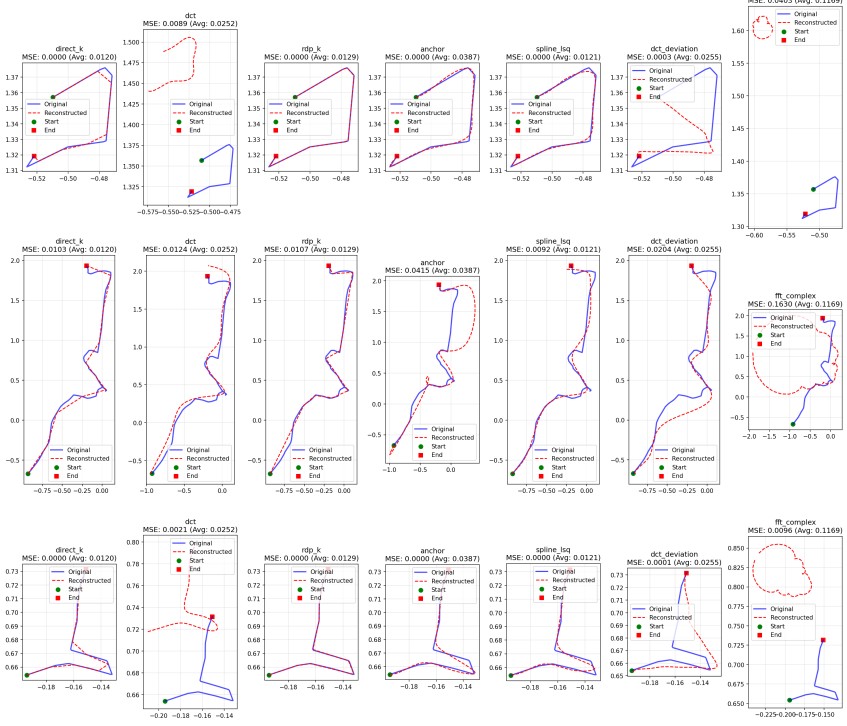

Figure 7: Parameter Number = 10

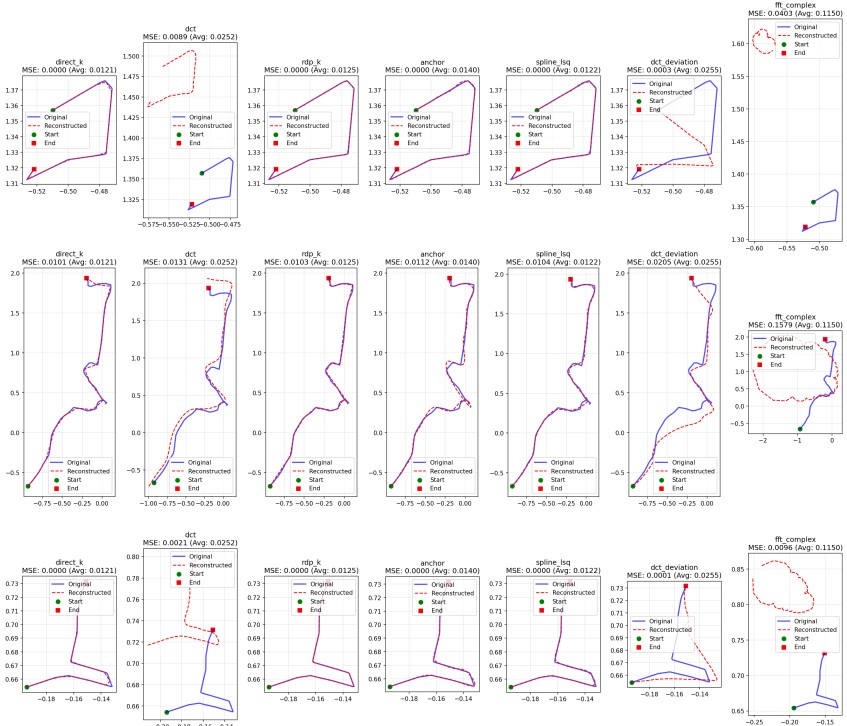

Figure 8: Parameter Number = 20

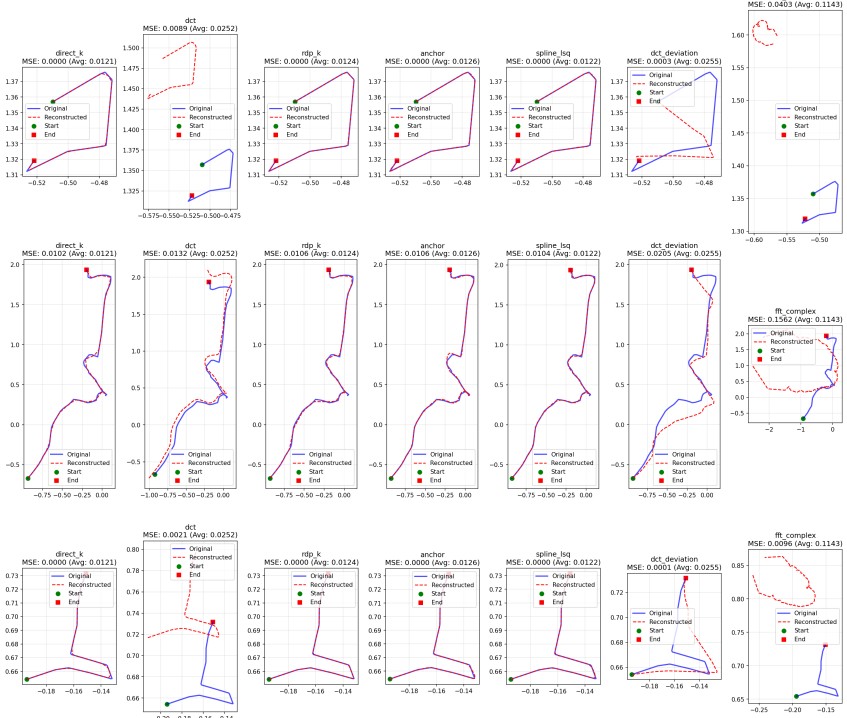

Figure 9: Parameter Number = 30

# B  ADDITIONAL COMPARISON WITH DENOISING DIFFUSION MODELS

One possible concern is that diffusion models might achieve comparable performance to *TrajFlow* if trained or sampled with a larger number of steps, rather than using the same limited steps as in flow matching. Our experiments confirm that increasing the number of steps indeed improves performance; for instance, with 300 steps, diffusion models approach the performance of the proposed method, although they remain slightly inferior (Table 3). Notably, while further increasing the number of steps might eventually surpass *TrajFlow*, this underscores the key advantage of our flow–matching–based approach: it achieves high fidelity while maintaining superior efficiency.

Table 3: TrajFlow-w/o-FM performance with different training/sampling steps (unit: km). Best metric in **bold**.

| Method | Density JS ↓ | $DTW_{med}$ ↓ | $Fr_{med}$ ↓ | $DTW_{IQR}$ ↓ | $DTW_{P10}$ ↓ | $DTW_{P90}$ ↓ | $Fr_{IQR}$ ↓ | $Fr_{P10}$ ↓ | $Fr_{P90}$ ↓ |
|---|---|---|---|---|---|---|---|---|---|
| *Central Tokyo* | | | | | | | | | |
| TrajFlow (ours) | **0.0674** | **20.350** | **0.304** | **13.392** | **10.574** | **39.119** | **0.174** | **0.200** | **0.674** |
| TrajFlow-w/o-FM (steps=10) | 0.3213 | 362.611 | 5.844 | 794.886 | 86.305 | 2096.804 | 14.045 | 1.149 | 37.590 |
| TrajFlow-w/o-FM (steps=50) | 0.2923 | 263.889 | 4.390 | 694.411 | 64.512 | 1827.793 | 12.156 | 0.855 | 32.903 |
| TrajFlow-w/o-FM (steps=100) | 0.2269 | 142.836 | 2.363 | 303.280 | 38.252 | 749.346 | 5.244 | 0.519 | 12.811 |
| TrajFlow-w/o-FM (steps=200) | 0.1208 | 36.434 | 0.583 | 49.489 | 15.035 | 130.881 | 0.975 | 0.247 | 2.437 |
| TrajFlow-w/o-FM (steps=300) | 0.0807 | 24.303 | 0.349 | 19.773 | 11.768 | 57.728 | 0.331 | 0.211 | 1.049 |
| *Tokyo Metropolis* | | | | | | | | | |
| TrajFlow (ours) | **0.1239** | **18.167** | **0.335** | **16.892** | **7.678** | **44.316** | **0.333** | **0.130** | **0.933** |
| TrajFlow-w/o-FM (steps=10) | 0.3833 | 470.857 | 7.668 | 994.283 | 154.963 | 3179.940 | 17.724 | 2.069 | 54.807 |
| TrajFlow-w/o-FM (steps=50) | 0.3657 | 429.968 | 6.866 | 951.462 | 115.913 | 2935.618 | 16.754 | 1.555 | 50.237 |
| TrajFlow-w/o-FM (steps=100) | 0.3351 | 344.806 | 5.774 | 781.324 | 84.420 | 2006.533 | 13.825 | 1.118 | 36.338 |
| TrajFlow-w/o-FM (steps=200) | 0.2600 | 139.901 | 2.423 | 163.029 | 45.480 | 416.262 | 3.040 | 0.644 | 7.104 |
| TrajFlow-w/o-FM (steps=300) | 0.1690 | 33.035 | 0.546 | 34.456 | 11.806 | 84.754 | 0.653 | 0.186 | 1.527 |
| *Japan* | | | | | | | | | |
| TrajFlow (ours) | **0.2270** | **10.977** | **0.192** | **18.221** | **3.984** | **55.964** | **0.361** | **0.072** | **1.119** |
| TrajFlow-w/o-FM (steps=10) | 0.5808 | 495.961 | 7.901 | 969.032 | 166.495 | 3267.659 | 17.900 | 2.276 | 56.966 |
| TrajFlow-w/o-FM (steps=50) | 0.5632 | 367.991 | 5.821 | 738.327 | 117.033 | 2585.896 | 13.551 | 1.512 | 45.142 |
| TrajFlow-w/o-FM (steps=100) | 0.5220 | 210.646 | 3.344 | 407.453 | 70.677 | 1332.320 | 7.386 | 0.996 | 23.968 |
| TrajFlow-w/o-FM (steps=200) | 0.3856 | 55.417 | 0.847 | 88.626 | 19.797 | 263.442 | 1.619 | 0.311 | 4.622 |
| TrajFlow-w/o-FM (steps=300) | 0.3047 | 26.125 | 0.412 | 36.912 | 9.376 | 113.981 | 0.700 | 0.151 | 2.094 |

Beyond the accuracy performance, we also compared the inference efficiency (time cost) between the proposed method and diffusion-based version (TrajFlow-w/o-FM). As Figure. 10 and Table. 4 shows, the flow matching-based approach has a much higher efficiency than the diffusion-based approach. Specifically, the diffusion-based version requires over 30× more inference time-cost while achieving accuracy levels that are comparable yet still lower than TrajFlow.

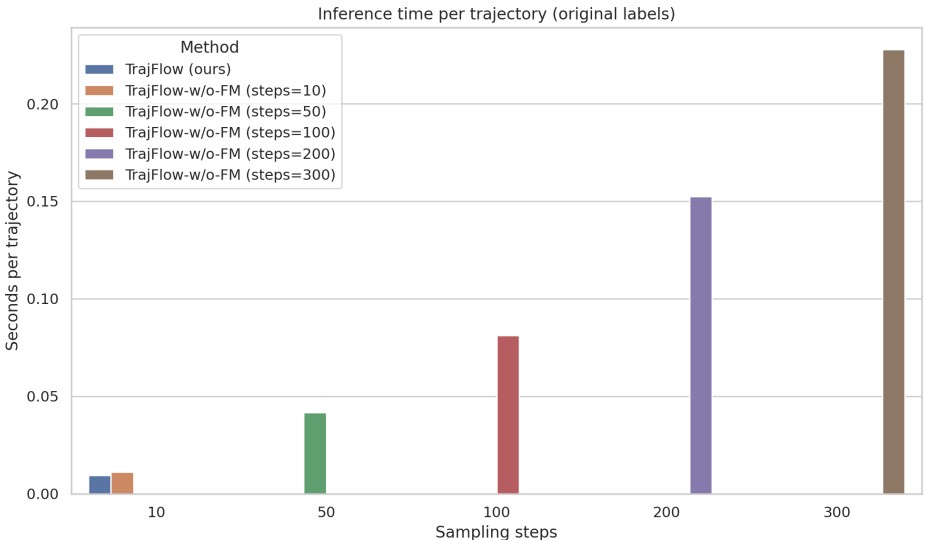

Figure 10: Bar plot for time cost comparison in inference stage.

Table 4: Comparison table for time cost in inference stage.

| Region | Method | Steps | Sec/Sample | Sampling Sec | Total Sec |
|---|---|---|---|---|---|
| Central Tokyo | TrajFlow | 10 | 0.0097 | 0.9696 | 4.8912 |
| Central Tokyo | TrajFlow-w/o-FM (steps=10) | 10 | 0.0100 | 1.0039 | 4.9646 |
| Central Tokyo | TrajFlow-w/o-FM (steps=50) | 50 | 0.0382 | 3.8235 | 7.7105 |
| Central Tokyo | TrajFlow-w/o-FM (steps=100) | 100 | 0.0761 | 7.6105 | 11.5326 |
| Central Tokyo | TrajFlow-w/o-FM (steps=200) | 200 | 0.1585 | 15.8546 | 19.8442 |
| Central Tokyo | TrajFlow-w/o-FM (steps=300) | 300 | 0.2237 | 22.3682 | 26.3010 |
| Tokyo Metropolis | TrajFlow | 10 | 0.0097 | 0.9735 | 4.8695 |
| Tokyo Metropolis | TrajFlow-w/o-FM (steps=10) | 10 | 0.0135 | 1.3465 | 5.7994 |
| Tokyo Metropolis | TrajFlow-w/o-FM (steps=50) | 50 | 0.0474 | 4.7377 | 8.6324 |
| Tokyo Metropolis | TrajFlow-w/o-FM (steps=100) | 100 | 0.0816 | 8.1608 | 12.1447 |
| Tokyo Metropolis | TrajFlow-w/o-FM (steps=200) | 200 | 0.1474 | 14.7445 | 18.6668 |
| Tokyo Metropolis | TrajFlow-w/o-FM (steps=300) | 300 | 0.2356 | 23.5593 | 27.5961 |
| Japan | TrajFlow | 10 | 0.0086 | 0.8575 | 4.7650 |
| Japan | TrajFlow-w/o-FM (steps=10) | 10 | 0.0095 | 0.9466 | 4.8438 |
| Japan | TrajFlow-w/o-FM (steps=50) | 50 | 0.0391 | 3.9072 | 7.8493 |
| Japan | TrajFlow-w/o-FM (steps=100) | 100 | 0.0860 | 8.6031 | 12.5301 |
| Japan | TrajFlow-w/o-FM (steps=200) | 200 | 0.1515 | 15.1521 | 19.0516 |
| Japan | TrajFlow-w/o-FM (steps=300) | 300 | 0.2240 | 22.4007 | 26.6460 |

## C TRANSPORTATION-MODE DIVERSITY

Figures 11 and Figure 12 compare the generated trajectories of our model with the ground truth in four modes of transport. Mode diversity is much more pronounced within Tokyo, where travelers frequently switch among walking, cycling, buses, subways, private cars, and multiple rail systems. In contrast, on nationwide trips, mobility patterns are dominated by a much smaller set of modes (primarily long-distance trains), while local segments are typically sparse or not captured at the same granularity.

### C.1 URBAN-WIDE(TOKYO)

In the ground-truth data, distinct patterns emerge: train trips typically span larger spatial scales and extend to the outskirts of Tokyo; car trips are also long but more concentrated in central areas; bike and walk trips both cluster within local communities, though with different levels of continuity. These diverse modal behaviors are well reproduced by our model, demonstrating its ability to capture transportation-mode heterogeneity.

### C.2 NATION-WIDE (JAPAN)

In the ground-truth data, we find that the train trajectories show more concentrate which is reasonable - the choices for long-distance railway network is limited, and car trajectories show more diverse routes. Most walk and bike trajectories are limited in local area.

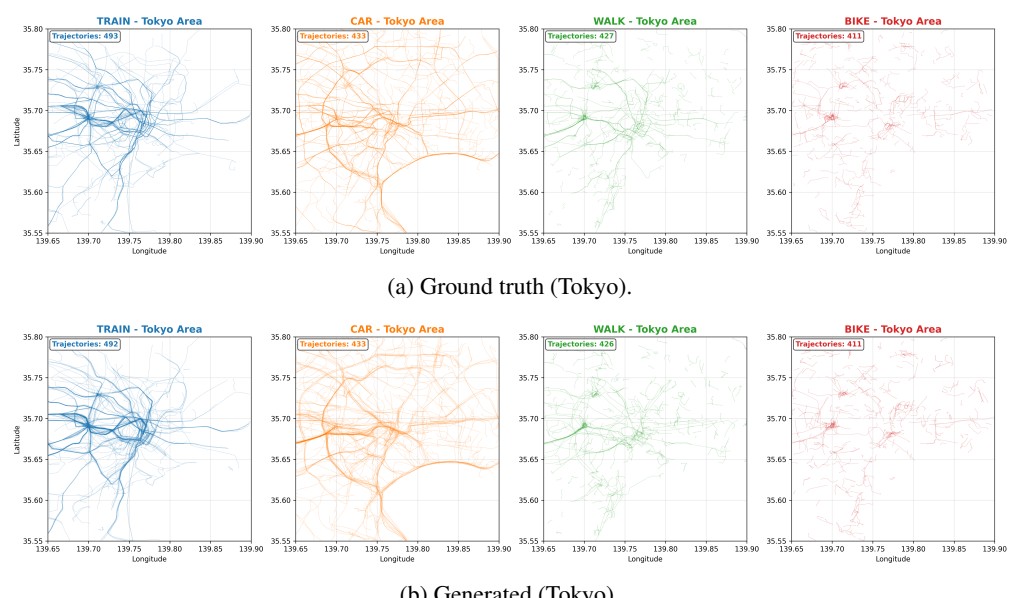

(a) Ground truth (Tokyo).

(b) Generated (Tokyo).

Figure 11: Transportation-mode diversity visualization: GT vs. Gen in Tokyo.

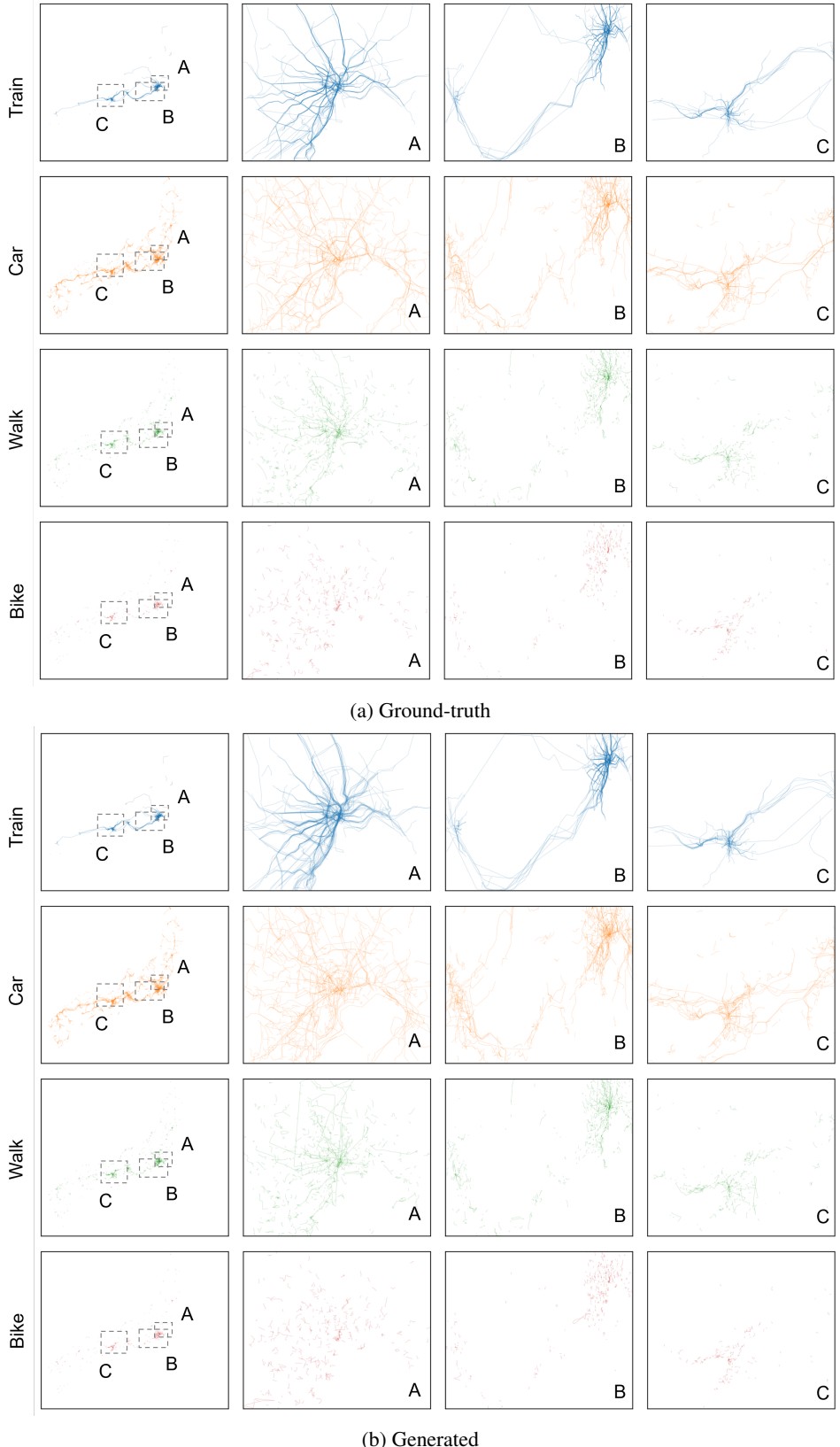

Figure 12: The capability to generate GPS trajectories under given transportation mode conditions. Ground-truth nation-wide trajectories shown with zoomed views highlighting three representative regions: (A) Tokyo Metropolis, (B) Tokaido Area, and (C) Kansai Area, across various transportation modes: Train, Car, Walk, Bike **(a)** shows the ground-truth trajectories. **(b)** shows the generated.

# D    REPRODUCIBILITY

## D.1    DATASET INFORMATION

In accordance with the data policy of Blogwatcher Inc., the complete ground-truth BlogWatcher dataset can only be accessed through a strict application process. Although the dataset used in our study originates from a single data provider, it is sourced from over 140 independent mobile applications (as noted on https://www.blogwatcher.co.jp/), ensuring substantial diversity in data collection. Because comparable multi-scale datasets are not publicly available, we are unable to benchmark TrajFlow on alternative nationwide or multi-level datasets. The dataset we used spans the period from January 2023 to December 2023 and contains approximately 3,000,000 (3 million) trajectories across Japan. For our experiments, we derive three subsets of different spatial scales: (1) Tokyo City, (2) Tokyo Metropolis, and (3) Nationwide.

| uid | segment_id | trans_mode1 | trans_mode2 | time | lat | lon |
|-----|-----------|-------------|-------------|------|-----|-----|
| . . . | 1 | STAY | STAY | 2023/2/6 0:00 | 35.9497 | 139.5576 |
| . . . | 1 | STAY | STAY | 2023/2/6 0:15 | 35.9497 | 139.5576 |
| . . . | . . . | . . . | . . . | . . . | . . . | . . . |
| . . . | 2 | MOVE | TRAIN | 2023/2/6 0:36 | 36.2269 | 114.1721 |
| . . . | . . . | . . . | . . . | . . . | . . . | . . . |
| . . . | 3 | MOVE | WALK | 2023/2/6 0:59 | 36.7236 | 114.4598 |

Table 5: Example rows of the dataset we used. (Note: this example data is only for showing data format without real lat/lon values.)

Each trajectory is pre-segmented according to transportation mode. The data fields relevant to this study are summarized in Table 5. Here, `uid` denotes an anonymized user identifier; however, we do not utilize any user-level identifiers in this work to avoid potential privacy concerns. The field `segment_id` indexes individual trajectory segments; `trans_mode1` distinguishes between `STAY` and `MOVE` states, and we use only the `MOVE` segments. The field `trans_mode2` provides fine-grained transportation modes such as `TRAIN`, `CAR`, `WALK`, and `BIKE`. Within movement segments, GPS points are recorded at an average interval of at least one minute.

To validate our method in open source dataset, we additionally evaluate on two open-source datasets in Chengdu and Xi'an, see Section E in the Appendix. In addition, a portion of the processed demo data, together with the implementation and generated results, will be made publicly available upon acceptance of this work.

## D.2    MODEL ARCHITECTURE

**Backbone:** We employ a U-Net Architecture as the vector field estimator.

- **Dimensions:** Base channels = 128. Channel multipliers = [1, 2, 2, 2].
- **Structure:** Two residual blocks per stage with SiLU activation. Attention mechanisms are applied at resolution 16.
- **Embeddings:** Timestep embedding dimension is 512 (4× base channels). Hidden dimension for condition embeddings is 512.

**Conditioning (Wide & Deep):** We utilize a "Wide & Deep" architecture to encode heterogeneous inputs:

- **Wide Component:** Linear projection of continuous features (e.g., speed, distance).
- **Deep Component:** Categorical embeddings for Departure Time (288 bins), Origin/Destination IDs, and Transportation Modes (5 classes).
- **Fusion:** The outputs are fused into a 128-dimensional condition vector, which is injected into the U-Net residual blocks via cross-attention or additive projection.

## D.3    TRAINING CONFIGURATION

- **Optimizer:** Adam optimizer ($\beta_1 = 0.9, \beta_2 = 0.999$) with no weight decay.

- **Learning Rate:** Initial learning rate set to $1 \times 10^{-4}$. We use a `ReduceLROnPlateau` scheduler (factor=0.5, patience=200) monitoring validation loss.

- **Batch Size:** 256 trajectories per GPU.

- **Training Budget:** Up to 500 epochs with Early Stopping (patience=5000 steps, delta=$1e^{-6}$).

- **Regularization:** We apply random condition dropout (Classifier-Free Guidance) during training to prevent overfitting and enable guidance during inference.

## D.4 ALGORITHM SETTINGS

- **Trajectory Harmonization (RDP):** We use the Ramer-Douglas-Peucker (RDP) algorithm for compression.
  - Target keypoints $M = 10$.
  - Epsilon tolerance determined via binary search ($\epsilon_{tol} = 1e^{-5}$).
  - Max sequence length $L_{max} = 120$.

- **Flow Matching Solver:**
  - **Method:** Euler ODE integrator.
  - **Inference Steps:** 10 steps (step size = 0.1). This setting was chosen to balance fidelity and efficiency (approx. 0.01s per sample).

- **Diffusion Baseline (for comparison):** Linear $\beta$ schedule ($1e^{-6}$ to $5e^{-2}$). DDIM sampler used with steps ranging from 10 to 300 for efficiency benchmarking.

## D.5 COMPUTING ENVIRONMENT

All experiments were implemented in **PyTorch** and executed on a cluster of four **NVIDIA RTX 6000 Ada GPUs** (40GB memory each). The software environment includes CUDA 12.6 and Python 3.9.

## E VALIDATION ON OPEN-SOURCE DATA

We report the results based on 1,000,000 trajectory samples for each city. For reference, the previous *DiffTraj* paper reported using 3,493,918 trajectories from Chengdu and 2,180,348 trajectories from Xi'an. Across both cities, *TrajFlow* maintains strong performance and exhibits trends consistent with our main results (see Table. 6).

Table 6: Evaluation grouped by region (unit = km). Best metric in **bold**.

| Method | Density JS ↓ | $DTW_{med}$ ↓ | $Fr_{med}$ ↓ | $DTW_{IQR}$ ↓ | $DTW_{P10}$ ↓ | $DTW_{P90}$ ↓ | $Fr_{IQR}$ ↓ | $Fr_{P10}$ ↓ | $Fr_{P90}$ ↓ |
|---|---|---|---|---|---|---|---|---|---|
| *Chengdu* | | | | | | | | | |
| TrajFlow (ours) | 0.211 | 35.911 | 0.547 | 29.418 | 15.839 | 72.048 | 0.392 | 0.279 | 0.959 |
| TrajFlow-w/o-OD | 0.0809 | 3.696 | 0.123 | 2.847 | 2.180 | 8.904 | 0.117 | 0.0614 | 0.362 |
| TrajFlow-w/o RDP | 0.206 | 34.687 | 0.540 | 30.020 | 14.650 | 69.999 | 0.383 | 0.266 | 0.957 |
| TrajFlow-w/o RDP & OD | **0.0140** | 2.158 | **0.0525** | **0.896** | 1.519 | 3.278 | 0.0243 | 0.0354 | 0.0880 |
| TrajFlow-w/o Flow | 0.208 | 36.457 | 0.561 | 30.017 | 15.948 | 73.523 | 0.397 | 0.278 | 0.975 |
| TrajFlow-w/o OD & Flow | 0.0811 | 3.724 | 0.123 | 2.800 | 2.142 | 8.844 | 0.119 | 0.0618 | 0.365 |
| TrajFlow-w/o RDP & Flow | 0.203 | 36.166 | 0.561 | 30.311 | 15.213 | 73.719 | 0.388 | 0.276 | 0.985 |
| DiffTraj (baseline) | 0.0224 | 3.524 | 0.0944 | 1.563 | 2.312 | 5.460 | 0.0482 | 0.0606 | 0.161 |
| *XiAn* | | | | | | | | | |
| TrajFlow (ours) | 0.284 | 43.457 | 0.564 | 35.390 | 16.948 | 79.586 | 0.379 | 0.277 | 0.897 |
| TrajFlow-w/o-OD | 0.0820 | 2.565 | 0.0842 | 2.009 | 1.530 | 8.488 | 0.116 | 0.0453 | 0.395 |
| TrajFlow-w/o RDP | 0.278 | 40.859 | 0.549 | 34.013 | 15.547 | 75.445 | 0.380 | 0.259 | 0.875 |
| TrajFlow-w/o RDP & OD | **0.0056** | 1.111 | **0.0279** | **0.401** | 0.805 | **1.611** | 0.0120 | 0.0192 | 0.0443 |
| TrajFlow-w/o Flow | 0.284 | 43.572 | 0.567 | 35.941 | 17.259 | 79.985 | 0.379 | 0.276 | 0.895 |
| TrajFlow-w/o OD & Flow | 0.0819 | 2.354 | 0.0833 | 2.115 | 1.387 | 8.162 | 0.119 | 0.0448 | 0.395 |
| TrajFlow-w/o RDP & Flow | 0.278 | 40.328 | 0.555 | 33.247 | 15.592 | 75.524 | 0.382 | 0.257 | 0.872 |
| DiffTraj (baseline) | 0.0070 | 1.154 | 0.0284 | 0.494 | **0.773** | 1.750 | 0.0132 | 0.0193 | 0.0466 |

## F    DISTINCTION FROM EXISTING APPROACHES

Our work introduces several key contributions that differentiate TrajFlow from existing trajectory-generation approaches:

- **First flow-matching model for GPS trajectory generation (methodological novelty).** Prior trajectory-generation methods—including recent diffusion-based models—rely on stochastic reverse-time SDE sampling and require tens to hundreds of denoising steps. In contrast, TrajFlow is the first model to apply the flow-matching paradigm to GPS trajectory generation. Moreover, we introduce a trajectory normalization scheme and a specialized architecture tailored for flow matching, enabling the model to better handle spatial heterogeneity and further improve generation performance.

- **First nationwide-scale, multi-geospatial-level GPS generator (problem-level novelty).** Existing models are restricted to small urban or single-city settings due to spatial heterogeneity and instability. TrajFlow is the first model evaluated at urban, metropolitan, and nationwide scales, trained on millions of mobile-phone GPS trajectories across Japan, demonstrating robust generalization across heterogeneous regions.

## G    PRACTICAL APPLICATION

TrajFlow handles variable-length trajectories through a conditioning–reconstruction strategy rather than asking the generative model to infer sequence length implicitly.

- Fixed-length representation during training (Section 4.4). For stable batch training under flow matching, all trajectories are represented using a fixed maximum length with padding and validity masks. This allows the model to learn continuous spatial dynamics without being affected by sequence-length variability.

- Explicit duration conditioning (Section 4.3). The model is conditioned on Travel Time and Departure Time, which provide explicit temporal context. Because trajectory duration is given as a conditioning variable, the model does not need to guess or estimate the temporal length.

- Length-consistent reconstruction at inference. At generation time, the model predicts the spatial shape of the trajectory (via harmonized RDP points). The final trajectory is then reconstructed to the target duration using the provided Travel Time condition, ensuring that generated samples match the desired temporal length—whether 10 minutes or 2 hours.

This strategy enables TrajFlow to robustly generate trajectories with diverse and accurate temporal lengths while retaining a stable training process.

## H  PRIVACY ISSUE

The inputs to our model are limited to departure time, origin–destination (OD) zones, and transportation mode, all of which are high-level, aggregated attributes that do not include or reveal any user-level identifiers (e.g., user IDs, device IDs, or fine-grained personal metadata). Therefore, the model is not exposed to information that could directly compromise individual privacy.

## I  MEMORIZATION RISKS

TrajFlow is designed to learn underlying mobility patterns (e.g., road network constraints, route choice) rather than memorizing specific coordinate sequences. Similar to diffusion/flow-matching models in computer vision—which learn the concept of an object rather than copying specific training images—our model generates trajectories by transforming Gaussian noise under OD and time conditions. The flow-matching process injects noise and learns a continuous probability flow from noise to data, making the generation inherently stochastic. As a result, each sample starts from a different noise seed, and even with identical OD/time conditions, the model produces diverse and novel trajectories rather than retrieving or replicating any stored instance. This stochasticity, as in models like Stable Diffusion, ensures diversity and prevents deterministic copying of training trajectories.

In addition, to explicitly prevent the model from "memorizing" specific conditional mappings (overfitting), we apply Classifier-Free Guidance (CFG) and dropout trick, which consists of a training regularization and an inference mechanism: Training (Condition Dropout): During training, we randomly mask the input conditions (OD/Time/Mode/etc.,) with a probability. This acts as a strong regularizer, forcing the model to learn the general, unconditional distribution of human mobility rather than relying on specific conditions to retrieve stored instances. Inference (Guidance Formula): During generation/inference, we compute a guided update using a weighted linear combination of the conditional and unconditional vector fields. This is the standard classifier-free guidance formula, which increases the model's sensitivity to the conditioning signal without causing the model to memorize specific condition–trajectory pairs. By adjusting the guidance weight, we can strengthen or relax condition adherence, enabling higher-quality and non-deterministic trajectory samples under the same conditions.

Third, empirically, we observed that the minimum DTW distances between generated samples and the training set are consistently non-zero. And even with the same condition, various routes could be generated.

## J  ETHICS STATEMENT

This work adheres to the ICLR Code of Ethics. All experiments were conducted on anonymized, large-scale mobile phone GPS trajectory data, which was used strictly in aggregated form and under strict privacy rules. No personally identifiable information (PII) or user-level attributes (e.g., age, gender, home–work identifiers, or persistent pseudonymous IDs) were accessed or utilized. Consequently, the model does not attempt to capture or infer individual preferences.

The proposed methods are designed to generate pseudo-GPS trajectories for research on mobility modeling and transportation systems, not to reconstruct or deanonymize individual user behavior. The use of RDP-based harmonization and OD-prediction modules is focused on improving computational efficiency and trajectory-level fidelity, without compromising privacy.

We acknowledge that trajectory data can be sensitive, and inappropriate applications may raise concerns around surveillance or discriminatory use. To mitigate this, we limit our study to methodological contributions and evaluation on aggregated data. Our scope is restricted to GPS trajectory generation (sequences of locations over time), rather than full human mobility modeling that would include activity semantics, trip-chain structure, or purpose-specific constraints.

We believe this research can benefit society by enabling scalable simulation tools for urban planning, transportation analysis, and disaster response, while respecting user privacy. All legal, ethical, and research integrity requirements have been followed in the preparation of this work.

