# OpenReview forum: "TrajFlow: Nation-wide Pseudo GPS Trajectory Generation with Flow Matching Models"
_ICLR.cc/2026/Conference — ICLR 2026 Poster_

### Official Review · Reviewer_vidk · 2025-10-26

**Soundness:** 3
**Presentation:** 3
**Contribution:** 3
**Rating:** 6
**Confidence:** 4

**Summary:**

This paper introduces TrajFlow, a flow-matching-based generative model for synthesizing pseudo-GPS trajectories at a nationwide scale. Addressing key limitations of diffusion-based methods—namely poor scalability, limited transportation-mode diversity, and high sampling cost—TrajFlow integrates a trajectory harmonization and reconstruction pipeline with conditional flow matching. It is evaluated on a large-scale mobile GPS dataset from Japan and demonstrated superior performance.

**Strengths:**

S1: It is the first application of flow matching to trajectory generation; novel integration of RDP-based harmonization and OD-conditioned normalization.

S2: Strong empirical performance across multiple scales and modes; comprehensive ablation studies validate design choices.

S3: Well-organized structure; clear problem motivation and contribution statement.

**Weaknesses:**

W1: The model is trained and evaluated only on Japanese data; it is unclear how well it generalizes to other countries with different urban structures or mobility patterns.

W2: While the paper claims efficiency gains, runtime and memory usage are not compared in detail with baselines (e.g., training time, GPU hours).

W3: Although privacy is discussed, the paper does not explore potential misuse scenarios (e.g., synthetic data used to infer real user behavior or re-identification risks).

W4: The evaluation focuses on Tokyo for mode diversity; nationwide mode-specific performance is not as thoroughly analyzed.

**Questions:**

Q1: How does TrajFlow perform on non-Japanese datasets, particularly in countries with less structured transportation networks?

Q2: Could the authors provide a more detailed comparison of training/inference time and memory usage versus diffusion-based models?

Q3: What are the potential risks of model misuse, and have any safeguards been considered to prevent re-identification or surveillance applications?

Q4: Why was RDP chosen over other curve simplification methods (e.g., spline fitting) in the final model, given that some alternatives performed similarly in the appendix?

---

> ### Author Response · Authors · 2025-11-19
> **Reply to Reviewer vidk**
>
> Dear reviewer vidk，
> Thank you for your time to evaluate our paper. We appreciate your acknowledging the novelty and performance of our framework. We would like to provide the following responses for your concern.
>
> **W1** Thanks for your comment, we have added addtional dataset in China, the strong performance on Chengdu and Xi'an confirms that TrajFlow generalizes well to cities with different transportation networks.(Please refer to the response in Q1).
>
> **W2**
> Thanks for your kind comment, to emperical vadicate it, we have measured the wall-clock time for generating batches of 100 trajectories on the same hardware.
>
> Utilizing an ODE solver with just 10 steps, TrajFlow generates a batch in approximately 0.97 seconds (avg. ~0.01s per sample). The diffusion-based approach (represented by TrajFlow-w/o-FM) requires significantly more steps to achieve comparable quality. With 300 steps, the generation takes approximately 22.4 seconds per batch. This corresponds to an observed speedup of roughly 23x compared to a 300-step diffusion model, making TrajFlow significantly more viable for large-scale simulations.
>
>
> **W3** Thank you for raising this important point. We agree that discussing potential misuse scenarios is essential for responsible deployment. While our submission focused primarily on technical contributions, we clarify our position and include an expanded discussion here. The inputs to our model are limited to departure time, origin–destination (OD) zones, and transportation mode, all of which are high-level, aggregated attributes that do not include or reveal any user-level identifiers (e.g., user IDs, device IDs, or fine-grained personal metadata). Therefore, the model is not exposed to information that could directly compromise individual privacy.
> We acknowledge that any mobility generative model could be misused if applied improperly. To mitigate this, we explicitly restrict TrajFlow’s use cases to research and planning applications, and we will document guidelines emphasizing that the model should not be used for user-level profiling or re-identification.
>
> **W4** Thank you for this insightful comment. We agree that mode-specific evaluation is valuable, and we clarify our design choice here. Mode diversity is much more pronounced within Tokyo, where travelers frequently switch among walking, cycling, buses, subways, private cars, and multiple rail systems. In contrast, on nationwide trips, mobility patterns are dominated by a much smaller set of modes (primarily long-distance trains and airplanes), while local segments are typically sparse or not captured at the same granularity. As a result, mode distribution outside Tokyo is highly skewed, and detailed per-mode analysis at the national scale becomes less meaningful and less comparable across regions.

---

> ### Author Response · Authors · 2025-11-19
> **Reply to Reviewer vidk (About questions)**
>
> **Q1** Thank you for raising this question. Our study focuses on learning multi-scale GPS trajectory distributions directly from raw coordinates, departure time, OD zones, and transportation mode—without using any road-network structure or map topology. Because TrajFlow does not require or depend on network graphs, lane geometry, or connectivity information, the method is not directly influenced by whether a city has highly structured or irregular transportation networks. For this reason, our main evaluation does not target datasets selected specifically based on network structure differences.
>
> Nevertheless, to examine generalizability beyond Japan, we applied TrajFlow to two open-source non-Japanese datasets—Chengdu and Xi’an in China—which feature more organic road layouts and less standardized mobility patterns compared to Japanese cities. Despite these differences, TrajFlow demonstrates stable and competitive performance, with trends consistent with those observed in Japan.
>
> #### **Chengdu**
>
> | Method | Density JS ↓ | DTW_med ↓ | Fr_med ↓ | DTW_IQR ↓ | DTW_P10 ↓ | DTW_P90 ↓ | Fr_IQR ↓ | Fr_P10 ↓ | Fr_P90 ↓ |
> |--------|---------------|-----------|----------|------------|------------|------------|-----------|-----------|-----------|
> | TrajFlow (ours) | 0.211 | 35.911 | 0.547 | 29.418 | 15.839 | 72.048 | 0.392 | 0.279 | 0.959 |
> | TrajFlow-w/o-OD | 0.0809 | 3.696 | 0.123 | 2.847 | 2.180 | 8.904 | 0.117 | 0.0614 | 0.362 |
> | TrajFlow-w/o RDP | 0.206 | 34.687 | 0.540 | 30.020 | 14.650 | 69.999 | 0.383 | 0.266 | 0.957 |
> | **TrajFlow-w/o RDP & OD** | **0.0140** | **2.158** | **0.0525** | **0.896** | **1.519** | **3.278** | **0.0243** | **0.0354** | **0.0880** |
> | TrajFlow-w/o Flow | 0.208 | 36.457 | 0.561 | 30.017 | 15.948 | 73.523 | 0.397 | 0.278 | 0.975 |
> | TrajFlow-w/o OD & Flow | 0.0811 | 3.724 | 0.123 | 2.800 | 2.142 | 8.844 | 0.119 | 0.0618 | 0.365 |
> | TrajFlow-w/o RDP & Flow | 0.203 | 36.166 | 0.561 | 30.311 | 15.213 | 73.719 | 0.388 | 0.276 | 0.985 |
> | DiffTraj (baseline) | 0.0224 | 3.524 | 0.0944 | 1.563 | 2.312 | 5.460 | 0.0482 | 0.0606 | 0.161 |
>
> ---
>
> #### **Xi'an**
>
> | Method | Density JS ↓ | DTW_med ↓ | Fr_med ↓ | DTW_IQR ↓ | DTW_P10 ↓ | DTW_P90 ↓ | Fr_IQR ↓ | Fr_P10 ↓ | Fr_P90 ↓ |
> |--------|---------------|-----------|----------|------------|------------|------------|-----------|-----------|-----------|
> | TrajFlow (ours) | 0.284 | 43.457 | 0.564 | 35.390 | 16.948 | 79.586 | 0.379 | 0.277 | 0.897 |
> | TrajFlow-w/o-OD | 0.0820 | 2.565 | 0.0842 | 2.009 | 1.530 | 8.488 | 0.116 | 0.0453 | 0.395 |
> | TrajFlow-w/o RDP | 0.278 | 40.859 | 0.549 | 34.013 | 15.547 | 75.445 | 0.380 | 0.259 | 0.875 |
> | **TrajFlow-w/o RDP & OD** | **0.0056** | **1.111** | **0.0279** | **0.401** | 0.805 | **1.611** | **0.0120** | **0.0192** | **0.0443** |
> | TrajFlow-w/o Flow | 0.284 | 43.572 | 0.567 | 35.941 | 17.259 | 79.985 | 0.379 | 0.276 | 0.895 |
> | TrajFlow-w/o OD & Flow | 0.0819 | 2.354 | 0.0833 | 2.115 | 1.387 | 8.162 | 0.119 | 0.0448 | 0.395 |
> | TrajFlow-w/o RDP & Flow | 0.278 | 40.328 | 0.555 | 33.247 | 15.592 | 75.524 | 0.382 | 0.257 | 0.872 |
> | DiffTraj (baseline) | 0.0070 | 1.154 | 0.0284 | 0.494 | **0.773** | 1.750 | 0.0132 | 0.0193 | 0.0466 |
>
> **Q2**  We have add the corresponding information in **Table 4 and Figure 8 in Appendix** in the manuscript of the new version.
>
> **Q3** Because TrajFlow does not operate on device IDs, user IDs, or continuous per-user trajectory histories, it cannot be used to re-create or track individuals. We explicitly restrict the intended use of TrajFlow to aggregate mobility research such as network-level planning, disaster response simulation, and policy evaluation. In the next version, we will add clear guidelines stating that the model should not be used for user profiling or fine-grained prediction of individual behavior.
>
> **Q4**  We chose RDP based on the empirical results in Table 2 (Appendix), which demonstrate that RDP provides the best trade-off between compression ratio (efficiency) and reconstruction fidelity. In addition, urban mobility trajectories are characterized by sharp turns (intersections) and straight segments. RDP is specifically designed to identify and preserve these critical inflection points. In contrast, Spline fitting tends to "oversmooth" sharp corners at lower parameter budgets, leading to higher geometric distortion as reflected in the higher DTW scores at K=5 and K=10.
>
> Therefore, RDP was selected to maximize spatial accuracy while keeping the sequence length short (K=10) to ensure the training and inference efficiency of the Flow Matching model.

---

### Official Review · Reviewer_P2bz · 2025-10-30

**Soundness:** 3
**Presentation:** 4
**Contribution:** 4
**Rating:** 6
**Confidence:** 5

**Summary:**

This paper presents a novel high-fidelity GPS trajectory generation model, called TrajFlow, which aims to address key challenges in current research. Although real GPS data is highly valuable, its application is hindered by privacy concerns, high costs, and access restrictions. Existing generation methods based on diffusion models have high fidelity but suffer from three major limitations: they are limited to small urban areas, lack diversity in multi-traffic patterns, and are inefficient in terms of training and inference. The TrajFlow applies the flow-matching paradigm to GPS trajectory generation. The core of the approach is to address the signal-to-noise ratio collapse problem encountered by diffusion models when scaled to large scales. It achieves this goal through a trajectory coordination and reconstruction strategy: the trajectories are first compressed using the RDP algorithm and then normalized to a uniform feature space for training. Experiments on a nationwide GPS dataset covering the entire country of Japan demonstrate that TrajFlow outperforms baselines, such as diffusion models, at city, metropolitan area, and nationwide scales, while maintaining the diversity of traffic patterns and being highly efficient.

**Strengths:**

1. This paper presents the first application of the flow-matching paradigm to the task of GPS trajectory generation, which addresses the problem that the performance of existing models, especially diffusion models, degrades dramatically when scaling from small urban scales to regional or national scales.
2. TrajFlow is much more efficient than the computationally expensive diffusion model, which requires a large number of sampling steps. TrajFlow achieves high fidelity in generation with only about 10 ODE steps.
3. TrajFlow overcomes the limitations of previous studies that are mainly limited to cab trajectory data, and is able to generate trajectories for multiple modes of transportation, including trains, cars, bicycles, and walking.

**Weaknesses:**

1. This paper lacks some details about the reproducibility of the models and algorithms, such as model architecture, parameter settings, code, algorithm process, etc.
2. This paper uses trajectory data with multiple travel modes and a national scale. However, it's not publicly available, and the authors do not present many details about the dataset. This limits the reader's ability to review the technical performance of this paper in depth.
3. The main evaluation metrics of the paper are biased towards space rather than time, such as DTW and Fréchet distance. These metrics are well-suited to measure the geometric similarity of two curves, but are limited in their ability to assess the temporal fidelity of trajectories.

**Questions:**

1. How does this paper deal with the problem of variable-length generation of data?
2. If the RDP algorithm is used to extract keypoints, it would hold the spatial characterization. So what is the difference between a trajectory generated using TrajFlow and one we interpolate directly based on keypoints?

---

> ### Author Response · Authors · 2025-11-19
> **Reply to Reviewer P2bz**
>
> Dear reviewer P2bz,
> Thank you for your time to evaluate our paper. We appreciate your acknowledging the novelty and performance of our framework. We would like to provide the following responses for your concern.
>
> **About reproducibility** We have enhanced this part in the new version manuscript.
>
> Current Details: Basic implementation details (e.g., PyTorch framework, NVIDIA H6000 GPUs, hyperparameters) are currently provided in **Appendix D**.
>
> Code Release: Upon acceptance, we will publicly release the source code, and generation scripts to facilitate further research.
>
> **About dataset** We acknowledge that nationwide, large-scale GPS trajectory datasets are extremely difficult to obtain. Although the dataset used in our study originates from a single data provider, it is sourced from over 140 independent mobile applications (as noted on https://www.blogwatcher.co.jp/), ensuring substantial diversity in data collection. Because comparable multi-scale datasets are not publicly available, we are unable to benchmark TrajFlow on alternative nationwide or multi-level datasets.
>
> However, to further validate the generalizability of our framework, we additionally evaluate TrajFlow on two open-source city-level datasets (Chengdu and Xi’an), which do not contain multi-scale structure but still provide meaningful evidence of cross-region robustness. Moreover, a portion of our processed demo data, along with the implementation and generated samples, will be released publicly upon acceptance of this work.
>
> The performance on the two open-source datasets is summarized below:
>
>   #### **Chengdu**
>
> | Method | Density JS ↓ | DTW_med ↓ | Fr_med ↓ | DTW_IQR ↓ | DTW_P10 ↓ | DTW_P90 ↓ | Fr_IQR ↓ | Fr_P10 ↓ | Fr_P90 ↓ |
> |--------|---------------|-----------|----------|------------|------------|------------|-----------|-----------|-----------|
> | TrajFlow (ours) | 0.211 | 35.911 | 0.547 | 29.418 | 15.839 | 72.048 | 0.392 | 0.279 | 0.959 |
> | TrajFlow-w/o-OD | 0.0809 | 3.696 | 0.123 | 2.847 | 2.180 | 8.904 | 0.117 | 0.0614 | 0.362 |
> | TrajFlow-w/o RDP | 0.206 | 34.687 | 0.540 | 30.020 | 14.650 | 69.999 | 0.383 | 0.266 | 0.957 |
> | **TrajFlow-w/o RDP & OD** | **0.0140** | **2.158** | **0.0525** | **0.896** | **1.519** | **3.278** | **0.0243** | **0.0354** | **0.0880** |
> | TrajFlow-w/o Flow | 0.208 | 36.457 | 0.561 | 30.017 | 15.948 | 73.523 | 0.397 | 0.278 | 0.975 |
> | TrajFlow-w/o OD & Flow | 0.0811 | 3.724 | 0.123 | 2.800 | 2.142 | 8.844 | 0.119 | 0.0618 | 0.365 |
> | TrajFlow-w/o RDP & Flow | 0.203 | 36.166 | 0.561 | 30.311 | 15.213 | 73.719 | 0.388 | 0.276 | 0.985 |
> | DiffTraj (baseline) | 0.0224 | 3.524 | 0.0944 | 1.563 | 2.312 | 5.460 | 0.0482 | 0.0606 | 0.161 |
>
> ---
>
> #### **Xi'an**
>
> | Method | Density JS ↓ | DTW_med ↓ | Fr_med ↓ | DTW_IQR ↓ | DTW_P10 ↓ | DTW_P90 ↓ | Fr_IQR ↓ | Fr_P10 ↓ | Fr_P90 ↓ |
> |--------|---------------|-----------|----------|------------|------------|------------|-----------|-----------|-----------|
> | TrajFlow (ours) | 0.284 | 43.457 | 0.564 | 35.390 | 16.948 | 79.586 | 0.379 | 0.277 | 0.897 |
> | TrajFlow-w/o-OD | 0.0820 | 2.565 | 0.0842 | 2.009 | 1.530 | 8.488 | 0.116 | 0.0453 | 0.395 |
> | TrajFlow-w/o RDP | 0.278 | 40.859 | 0.549 | 34.013 | 15.547 | 75.445 | 0.380 | 0.259 | 0.875 |
> | **TrajFlow-w/o RDP & OD** | **0.0056** | **1.111** | **0.0279** | **0.401** | 0.805 | **1.611** | **0.0120** | **0.0192** | **0.0443** |
> | TrajFlow-w/o Flow | 0.284 | 43.572 | 0.567 | 35.941 | 17.259 | 79.985 | 0.379 | 0.276 | 0.895 |
> | TrajFlow-w/o OD & Flow | 0.0819 | 2.354 | 0.0833 | 2.115 | 1.387 | 8.162 | 0.119 | 0.0448 | 0.395 |
> | TrajFlow-w/o RDP & Flow | 0.278 | 40.328 | 0.555 | 33.247 | 15.592 | 75.524 | 0.382 | 0.257 | 0.872 |
> | DiffTraj (baseline) | 0.0070 | 1.154 | 0.0284 | 0.494 | **0.773** | 1.750 | 0.0132 | 0.0193 | 0.0466 |
>
> We provide more detail about this data here and will also update the appendix in the next version.
>
> **About evaluation metrics**
>
> Implicit Temporal Constraints: Our model is explicitly conditioned on Departure Time and Travel Time (Section 4.3). The generation process creates trajectory points that inherently respect these temporal constraints. If not, it will be punlished by the DTW: As defined in Eq. 6, DTW explicitly aligns sequences in the time dimension. It penalizes trajectories that are spatially similar but temporally mismatched (e.g., wrong speed profiles).

---

> ### Author Response · Authors · 2025-11-19
> **Reply to Reviewer P2bz (about questions)**
>
> **Variable-length generation**
> Thank you for the question. TrajFlow handles variable-length trajectories through a conditioning–reconstruction strategy rather than asking the generative model to infer sequence length implicitly.
>
> 1. Fixed-length representation during training (Section 4.4).
> For stable batch training under flow matching, all trajectories are represented using a fixed maximum length with padding and validity masks. This allows the model to learn continuous spatial dynamics without being affected by sequence-length variability.
>
> 2. Explicit duration conditioning (Section 4.3).
> The model is conditioned on Travel Time and Departure Time, which provide explicit temporal context. Because trajectory duration is given as a conditioning variable, the model does not need to guess or estimate the temporal length.
>
> 3. Length-consistent reconstruction at inference.
> At generation time, the model predicts the spatial shape of the trajectory (via harmonized RDP points). The final trajectory is then reconstructed to the target duration using the provided Travel Time condition, ensuring that generated samples match the desired temporal length—whether 10 minutes or 2 hours.
>
> This strategy enables TrajFlow to robustly generate trajectories with diverse and accurate temporal lengths while retaining a stable training process.
>
> **Usage of RDP algorithm**
> Thank you for the question. We would like to clarify how TrajFlow uses RDP and why the generated trajectories differ from simply interpolating RDP keypoints.
>
> First, TrajFlow does not use RDP keypoints as input. The model learns to generate the RDP-compressed trajectory representation directly from Gaussian noise, conditioned on OD and temporal information. In contrast, interpolation requires RDP keypoints that already come from a real trajectory. TrajFlow produces these keypoints itself from a learned distribution, meaning the generated spatial structure is new rather than extracted from an existing trajectory. Interpolation is only the final decoding step to map the generated compact representation back into geographic space.
>
> Second, RDP is used only as a dimensionality-reduction method. We compress each raw trajectory (often around 120 points) into a smaller set of keypoints (around 10). This compact representation greatly improves flow-matching training efficiency and stability, especially across multiple spatial scales.
>
> Finally, we chose RDP because it provides the best balance between compression ratio and reconstruction accuracy. As shown in our ablation study, RDP achieves lower DTW and Fréchet distances than alternatives such as DCT, spline fitting, or FFT. This ensures that the simplified representation maintains high spatial fidelity before the interpolation step.
>
> In summary, TrajFlow does not rely on pre-existing RDP keypoints. Instead, it learns to generate new spatial keypoints, while RDP serves only as an efficient representation during training.

---

### Official Review · Reviewer_577e · 2025-11-01

**Soundness:** 2
**Presentation:** 2
**Contribution:** 2
**Rating:** 4
**Confidence:** 3

**Summary:**

This paper addresses the problem of pseudo-GPS trajectory generation, which is challenged by issues of spatial scalability, multi-modal transportation diversity, and generation efficiency. To tackle these limitations, the authors propose TrajFlow, a novel flow-matching-based generative framework that incorporates trajectory harmonization and reconstruction within a conditional generative paradigm. Experimental results demonstrate the effectiveness of the proposed method.

**Strengths:**

S1. This paper studies the problem of pseudo GPS trajectory generation, which seems interesting.

S2. The paper presents the first flow-matching-based generative framework.

S3. Experiments show that the proposed TrajFlow outperforms the existing baselines.

**Weaknesses:**

W1. Novelty: The paper would benefit from a deeper discussion clarifying the differences between the proposed approach and more recent baselines.

W2. Datasets: Experiments are conducted on only one dataset, which limits the generalizability of the conclusions. It is recommended to include additional commonly used datasets such as Chengdu and Xi’an to strengthen the empirical validation.

W3.Baseline: The baselines used for comparison (from 2020, 2021, and 2023) are relatively outdated. The paper should include more recent baselines mentioned in the related work section and existing work such as Diffusion-TS to ensure a fair and comprehensive comparison.
[1] Interpretable Diffusion for General Time Series Generation, ICLR 2024.

W4. Lack of Complexity Analysis: The paper does not provide a theoretical analysis of time and space complexity. Including such analysis would help readers better understand the computational efficiency and scalability of the proposed framework.

W5. Reproducibility: The paper lacks the codes, which may hinder reproducibility.

**Questions:**

Q1: Missing Figure and Table References: Some figures (e.g., Figures 1 and 3) and tables (e.g., Table 1) are not referenced, which affects readability.

Q2: ODE is missing citation.

Q3: The methodology section is somewhat difficult to follow. For instance, it is unclear where is  Figure 3 referred and how it aligns with the overall model design. It is better to give more discussions.

Q4: Why are these evaluation metrics chosen? What is the rationale for using P10/P90 to describe central accuracy and dispersion?

---

> ### Author Response · Authors · 2025-11-19
> **Reply to Reviewer 577e**
>
> Dear reviewer 577e,
> Thank you for your effort in reviewing our paper. We appreciate recognizing the meaning of our work and experimental results. We would like to address your comments in detail as follows:
>
> **W1** Our work introduces several key contributions that differentiate TrajFlow from existing trajectory-generation approaches:
> 1. **First flow-matching model for GPS trajectory generation (methodological novelty**.
> Prior trajectory-generation methods—including recent diffusion-based models—rely on stochastic reverse-time SDE sampling and require tens to hundreds of denoising steps. In contrast, TrajFlow is the first model to apply the flow-matching paradigm to GPS trajectory generation. Moreover, we introduce a trajectory normalization scheme and a specialized architecture tailored for flow matching, enabling the model to better handle spatial heterogeneity and further improve generation performance.
> 2. **First nationwide-scale, multi-geospatial-level GPS generator (problem-level novelty**.
> Existing models are restricted to small urban or single-city settings due to spatial heterogeneity and instability. TrajFlow is the first model evaluated at urban, metropolitan, and nationwide scales, trained on millions of mobile-phone GPS trajectories across Japan, demonstrating robust generalization across heterogeneous regions.
>
> **W2** Thank you for your comment. Our primary motivation is to study multi-scale GPS trajectory generation under the flow-matching paradigm, which is why the main paper focuses on Tokyo (urban), Kanto (metropolitan), and nationwide scales. These scales allow us to evaluate TrajFlow on spatial heterogeneity, multi-level generalization of our problem formulation.
>
> Nevertheless, we agree that validating on additional city-scale datasets can further strengthen empirical generalizability. Following your suggestion, we conducted experiments on the Chengdu and Xi’an city-level datasets. Due to time constraints, we report here the results based on 1,000,000 trajectories data for each city (the previous DiffTraj paper reported that they used 3,493,918 trajectories of Chengdu and 2,180,348 trajectories of Xi'an). Across both cities, TrajFlow maintains strong performance and shows consistent trends with our main results.
>
> #### **Chengdu**
>
> | Method | Density JS ↓ | DTW_med ↓ | Fr_med ↓ | DTW_IQR ↓ | DTW_P10 ↓ | DTW_P90 ↓ | Fr_IQR ↓ | Fr_P10 ↓ | Fr_P90 ↓ |
> |--------|---------------|-----------|----------|------------|------------|------------|-----------|-----------|-----------|
> | TrajFlow (ours) | 0.211 | 35.911 | 0.547 | 29.418 | 15.839 | 72.048 | 0.392 | 0.279 | 0.959 |
> | TrajFlow-w/o-OD | 0.0809 | 3.696 | 0.123 | 2.847 | 2.180 | 8.904 | 0.117 | 0.0614 | 0.362 |
> | TrajFlow-w/o RDP | 0.206 | 34.687 | 0.540 | 30.020 | 14.650 | 69.999 | 0.383 | 0.266 | 0.957 |
> | **TrajFlow-w/o RDP & OD** | **0.0140** | **2.158** | **0.0525** | **0.896** | **1.519** | **3.278** | **0.0243** | **0.0354** | **0.0880** |
> | TrajFlow-w/o Flow | 0.208 | 36.457 | 0.561 | 30.017 | 15.948 | 73.523 | 0.397 | 0.278 | 0.975 |
> | TrajFlow-w/o OD & Flow | 0.0811 | 3.724 | 0.123 | 2.800 | 2.142 | 8.844 | 0.119 | 0.0618 | 0.365 |
> | TrajFlow-w/o RDP & Flow | 0.203 | 36.166 | 0.561 | 30.311 | 15.213 | 73.719 | 0.388 | 0.276 | 0.985 |
> | DiffTraj (baseline) | 0.0224 | 3.524 | 0.0944 | 1.563 | 2.312 | 5.460 | 0.0482 | 0.0606 | 0.161 |
>
> ---
>
> #### **Xi'an**
>
> | Method | Density JS ↓ | DTW_med ↓ | Fr_med ↓ | DTW_IQR ↓ | DTW_P10 ↓ | DTW_P90 ↓ | Fr_IQR ↓ | Fr_P10 ↓ | Fr_P90 ↓ |
> |--------|---------------|-----------|----------|------------|------------|------------|-----------|-----------|-----------|
> | TrajFlow (ours) | 0.284 | 43.457 | 0.564 | 35.390 | 16.948 | 79.586 | 0.379 | 0.277 | 0.897 |
> | TrajFlow-w/o-OD | 0.0820 | 2.565 | 0.0842 | 2.009 | 1.530 | 8.488 | 0.116 | 0.0453 | 0.395 |
> | TrajFlow-w/o RDP | 0.278 | 40.859 | 0.549 | 34.013 | 15.547 | 75.445 | 0.380 | 0.259 | 0.875 |
> | **TrajFlow-w/o RDP & OD** | **0.0056** | **1.111** | **0.0279** | **0.401** | 0.805 | **1.611** | **0.0120** | **0.0192** | **0.0443** |
> | TrajFlow-w/o Flow | 0.284 | 43.572 | 0.567 | 35.941 | 17.259 | 79.985 | 0.379 | 0.276 | 0.895 |
> | TrajFlow-w/o OD & Flow | 0.0819 | 2.354 | 0.0833 | 2.115 | 1.387 | 8.162 | 0.119 | 0.0448 | 0.395 |
> | TrajFlow-w/o RDP & Flow | 0.278 | 40.328 | 0.555 | 33.247 | 15.592 | 75.524 | 0.382 | 0.257 | 0.872 |
> | DiffTraj (baseline) | 0.0070 | 1.154 | 0.0284 | 0.494 | **0.773** | 1.750 | 0.0132 | 0.0193 | 0.0466 |
>
> We have also included these extended results in the **Appendix E** in the manuscript of the new version.

---

> ### Author Response · Authors · 2025-11-19
> **Continue reply to the weakness**
>
> **W3** We appreciate the suggestion to consider Diffusion-TS (Yuan et al., ICLR 2024). We have closely examined this work and identified fundamental incompatibilities with the trajectory generation task:
>
>  a) Incompatibility of the "Seasonal-Trend" Mechanism: The core innovation of Diffusion-TS is its "Seasonal-Trend Decomposition" architecture , which explicitly uses Fourier synthetic layers  to model periodic patterns. This is highly effective for periodic signals (e.g., electricity load). However, a single GPS trajectory is inherently aperiodic; it represents a spatial transition from Origin to Destination, not a recurrent oscillation. Forcing Fourier-based decomposition on spatial paths introduces incorrect modeling assumptions.
>
>   b) The difenition are also different: Crucially, TrajFlow does not treat trajectory generation as a standard time-series regression problem on a fixed temporal grid. Unlike Diffusion-TS which models values over fixed intervals, our framework treats temporal attributes as synthesized variables, derived from the conditional Departure Time, Travel Time, and the generated spatial waypoints. This formulation allows TrajFlow to reconstruct continuous spatial paths rather than just regressing point-wise values at rigid timestamps, ensuring higher fidelity in reproducing mobility dynamics.
>
>   Though we want to use more powerful SOTA model as baselines, we have to say that human mobility generation is not a very hot topic and the DiffTraj (Zhu et al., 2023) and its variant is indeed the SOTA model we could compare with.
>
>
> **W4** We have added the complexity analysis as follows:
>
>   a) Time Complexity (Major): This is a major advantage of our Flow Matching approach. Standard Diffusion models (e.g., DiffTraj) typically require 300\~500 denoising steps to generate high-quality samples (even trick like DDIM sampling be used, at least 100~200 steps is necessary). In contrast, TrajFlow utilizes a Continuous Normalizing Flow (CNF) with an ODE solver, which allows for "straighter" generation paths. It achieves high-fidelity results with only 10–20 ODE solver steps (using RK4 or Euler). This results in an inference speedup of approximately 20x–50x, making it significantly more scalable for large-scale simulation.
>   b) Space Complexity (Minor): The memory consumption is $O(L \cdot D)$, where $L$ is the trajectory length and $D$ is the feature dimension. In our nationwide model, the RDP compression effectively reduces $L$, ensuring the model remains memory-efficient even for long-distance inter-prefectural trips.
>
> **W5** A portion of the processed demo data, together with the implementation and generated results, will be made publicly available upon acceptance of this work.

---

> ### Author Response · Authors · 2025-11-19
> **Reply to Reviewer 577e (about questions)**
>
> **Q1-Q2** Thanks for pointing out this issues. We have fixed them at the new version.
>
> **Q3** Thank you for this comment. Figure 3 serves as an overview of Section 4, summarizing not only the proposed design and architecture but also the training and generation stages of our framework. We consolidated these components into a single figure to present a coherent view of our study.
> To clarify the correspondence: the upper-left portion of Figure 3 illustrates the model design described in Section 4.2, the upper-right portion corresponds to the training process in Section 4.3, and the lower portion visualizes the generation stage discussed in Section 4.4. We agree that this linkage should be stated more clearly, and we will revise the text in the next version to make these connections explicit.
>
> **Q4** There are two reasons why we used metrics with DTW/Frechet and also the P10/p90:
>
> 1.  Existing baselines (e.g., the SOTA model DiffTraj ) primarily utilize statistical metrics such as JS-divergence of aggregated OD, JS-divergence of aggregated trajectory density. These metrics assess whether the population statistics match the real data but fail to quantify whether individual generated paths are geometrically realistic.  While we retain Density JS to ensure population-level consistency, we explicitly introduced DTW and Fréchet Distance. These are critical for measuring geometric and temporal fidelity at the trajectory level (e.g., smoothness, route choice, and shape alignment), ensuring the model generates realistic movement paths rather than just statistically correct endpoints.
>
> 2. Rationale for P10/P90 (Robustness in Long-tailed Distributions): We report P10 and P90 alongside the median to capture the model's robustness across the long-tailed complexity of human mobility: human mobility is usually long-tail, reporting only the Median (Central Accuracy) may mask model failures on complex outliers. By reporting P10/P90 (extreme/long-tail case), we demonstrate that TrajFlow remains stable and accurate even for the long-tail part trajectories, rather than just fitting the easy majority.

---

### Official Review · Reviewer_CQzo · 2025-11-01

**Soundness:** 4
**Presentation:** 4
**Contribution:** 4
**Rating:** 10
**Confidence:** 5

**Summary:**

The paper proposes a new generative model for GPS trajectories of human mobility. The architecture is based on flow-matching. The GPS trajectories are first normalized and then simplified to both help with efficiency and training stability. The model is shown to outperform state-of-the-art models across both trajectory-level and aggregate-level evaluation measures.

**Strengths:**

1. Very strong performance for trajectory generation at nation-level scale
2. The ablation study not only shows the importance of each part, but also discusses some of the inherent limitations of using a global coordinate frame when generating trajectories, which is that it introduces the risk of small details being lost when different trajectories have different scales.
3. Provides new insights into what is important for generating trajectories: trajectory simplification, normalization, and flow matching are all critical components.

**Weaknesses:**

1. All results are based on a single dataset, which is not publicly accessible.
2. Auxiliary data is required to sample from the model: departure times, OD pairs, and transportation modes.
3. Limited discussion around the risk of memorization. The DTW measure shows the average DTW distance to the closest real trajectory. While achieving a low score might seem positive, it could also indicate that the model has learned to copy the training set, potentially increasing the risk of leaking private information. This can also lead to inflated evaluation scores, as copying training data can improve evaluation metrics without the model genuinely learning to generate novel trajectories.

**Questions:**

1. How are departure times, OD pairs, and transportation modes obtained in practice?
2. Could you elaborate on why TrajFlow-w/o RDB & OD shows strong performance on the Central Tokyo region?
3. Is there a risk that the model has memorized the training set? The reported DTW is the average across multiple generated trajectories, but what is the smallest value you have observed for individual trajectories?

---

> ### Author Response · Authors · 2025-11-19
> **Reply to Reviewer CQzo**
>
> Dear Reviewer CQzo,
> Thank you sincerely for taking your valuable time to provide constructive feedback on this paper. We really appreciate the reviewer for the positive feedback. We would like to address your comments in detail as follows:
>
> #### Weakness
> **About dataset** We acknowledge that nationwide, large-scale GPS trajectory datasets are extremely difficult to obtain. Although the dataset used in our study originates from a single data provider, it is sourced from over 140 independent mobile applications (as noted on https://www.blogwatcher.co.jp/), ensuring substantial diversity in data collection. Because comparable multi-scale datasets are not publicly available, we are unable to benchmark TrajFlow on alternative nationwide or multi-level datasets.
>
> However, to further validate the generalizability of our framework, we additionally evaluate TrajFlow on two open-source urban-level datasets (Chengdu and Xi’an), which do not contain multi-scale structure (it's urban-scale only) but still provide meaningful evidence of cross-region robustness. Moreover, a portion of our processed demo data, along with the implementation and generated samples, will be released publicly upon acceptance of this work.
>
> The performance on the two open-source datasets is summarized below:
>
>   #### **Chengdu**
>
> | Method | Density JS ↓ | DTW_med ↓ | Fr_med ↓ | DTW_IQR ↓ | DTW_P10 ↓ | DTW_P90 ↓ | Fr_IQR ↓ | Fr_P10 ↓ | Fr_P90 ↓ |
> |--------|---------------|-----------|----------|------------|------------|------------|-----------|-----------|-----------|
> | TrajFlow (ours) | 0.211 | 35.911 | 0.547 | 29.418 | 15.839 | 72.048 | 0.392 | 0.279 | 0.959 |
> | TrajFlow-w/o-OD | 0.0809 | 3.696 | 0.123 | 2.847 | 2.180 | 8.904 | 0.117 | 0.0614 | 0.362 |
> | TrajFlow-w/o RDP | 0.206 | 34.687 | 0.540 | 30.020 | 14.650 | 69.999 | 0.383 | 0.266 | 0.957 |
> | **TrajFlow-w/o RDP & OD** | **0.0140** | **2.158** | **0.0525** | **0.896** | **1.519** | **3.278** | **0.0243** | **0.0354** | **0.0880** |
> | TrajFlow-w/o Flow | 0.208 | 36.457 | 0.561 | 30.017 | 15.948 | 73.523 | 0.397 | 0.278 | 0.975 |
> | TrajFlow-w/o OD & Flow | 0.0811 | 3.724 | 0.123 | 2.800 | 2.142 | 8.844 | 0.119 | 0.0618 | 0.365 |
> | TrajFlow-w/o RDP & Flow | 0.203 | 36.166 | 0.561 | 30.311 | 15.213 | 73.719 | 0.388 | 0.276 | 0.985 |
> | DiffTraj (baseline) | 0.0224 | 3.524 | 0.0944 | 1.563 | 2.312 | 5.460 | 0.0482 | 0.0606 | 0.161 |
>
> ---
>
> #### **Xi'an**
>
> | Method | Density JS ↓ | DTW_med ↓ | Fr_med ↓ | DTW_IQR ↓ | DTW_P10 ↓ | DTW_P90 ↓ | Fr_IQR ↓ | Fr_P10 ↓ | Fr_P90 ↓ |
> |--------|---------------|-----------|----------|------------|------------|------------|-----------|-----------|-----------|
> | TrajFlow (ours) | 0.284 | 43.457 | 0.564 | 35.390 | 16.948 | 79.586 | 0.379 | 0.277 | 0.897 |
> | TrajFlow-w/o-OD | 0.0820 | 2.565 | 0.0842 | 2.009 | 1.530 | 8.488 | 0.116 | 0.0453 | 0.395 |
> | TrajFlow-w/o RDP | 0.278 | 40.859 | 0.549 | 34.013 | 15.547 | 75.445 | 0.380 | 0.259 | 0.875 |
> | **TrajFlow-w/o RDP & OD** | **0.0056** | **1.111** | **0.0279** | **0.401** | 0.805 | **1.611** | **0.0120** | **0.0192** | **0.0443** |
> | TrajFlow-w/o Flow | 0.284 | 43.572 | 0.567 | 35.941 | 17.259 | 79.985 | 0.379 | 0.276 | 0.895 |
> | TrajFlow-w/o OD & Flow | 0.0819 | 2.354 | 0.0833 | 2.115 | 1.387 | 8.162 | 0.119 | 0.0448 | 0.395 |
> | TrajFlow-w/o RDP & Flow | 0.278 | 40.328 | 0.555 | 33.247 | 15.592 | 75.524 | 0.382 | 0.257 | 0.872 |
> | DiffTraj (baseline) | 0.0070 | 1.154 | 0.0284 | 0.494 | **0.773** | 1.750 | 0.0132 | 0.0193 | 0.0466 |
>
>
> **About practical application**  Thank you for the comment. We would like to clarify that the goal of this study is GPS trajectory generation, which specifically focuses on modeling the fine-grained “where and how” aspects of human movement (i.e., detailed paths and transportation modes). In this formulation, the model is conditioned on departure time, OD zones, and transportation mode—all of which are high-level contextual variables that are commonly available in mobility studies and do not contain personal identifiers.
>
> **About memorization risk** Please check the following response to **Question**.

---

> ### Author Response · Authors · 2025-11-19
> **Reply to Reviewer CQzo (about questions)**
>
> **About pratical application (Condition Data Acquisition)**  Thank you for this practical question. In real-world deployment, these conditional inputs (Departure Time, OD, Mode) are readily available from standard regional travel surveys (e.g., Person Trip surveys) or census data, which are routinely collected by transportation agencies.
>
> 1.  Obtaining these aggregate-level statistics is relatively low-cost and poses minimal privacy risks compared to collecting continuous individual GPS logs. As shown in seminal privacy studies, raw trajectory data carries a high risk of re-identification, whereas our framework relies only on non-sensitive, aggregated condition vectors.
>
> 2. Our model serves as a powerful simulation tool that bridges the gap between coarse-grained statistical data and fine-grained mobility analysis. It aligns with the standard pipeline of agent-based simulations, allowing planners to take easy-to-obtain survey data and "upsample" it into realistic, high-fidelity GPS trajectories for detailed infrastructure planning, without needing to track real individuals.
>
> **About performance**   It is a meaningful observation. The RDP compression and OD prediction modules were designed specifically to handle nationwide spatial heterogeneity and multi-scale complexity. While RDP is essential for efficiency on a national scale, it becomes unnecessary in the small, high-density Central Tokyo region (20km x 20km).
>   In this localized setting, RDP tends to smooth out fine-grained details that are critical for city-level metrics, and the global guidance from OD prediction becomes redundant for short trips. Therefore, TrajFlow-w/o RDP & OD performs better locally because it generates raw coordinates directly, preserving those details. This is not a problem, in contrast, this actually highlights the adaptability of our framework: the full architecture is necessary for multi-scale/national tasks, while the lighter backbone is sufficient for city-level generation.
>
> **About memorization**  Firstly, in terms of concept descrimination. We think we should descriminate between learning mobility patterns and memorizing instances. TrajFlow is designed to learn underlying mobility patterns (e.g., road network constraints, route choice) rather than memorizing specific coordinate sequences. Similar to diffusion models in computer vision—which learn the concept of an object rather than copying specific training images—our model generates trajectories by transforming Gaussian noise under OD and time conditions. The flow-matching process injects noise and learns a continuous probability flow from noise to data, making the generation inherently stochastic. As a result, each sample starts from a different noise seed, and even with identical OD/time conditions, the model produces diverse and novel trajectories rather than retrieving or replicating any stored instance. This stochasticity, as in models like Stable Diffusion, ensures diversity and prevents deterministic copying of training trajectories.
>
>   In addition, to explicitly prevent the model from "memorizing" specific conditional mappings (overfitting), we apply Classifier-Free Guidance (CFG) and dropout trick, which consists of a training regularization and an inference mechanism: Training (Condition Dropout): During training, we randomly mask the input conditions (OD/Time) with a probability $p_{drop}$. This acts as a strong regularizer, forcing the model to learn the general, unconditional distribution of human mobility rather than relying on specific conditions to retrieve stored instances. Inference (Guidance Formula): During generation, we perform sampling using the linear combination $v_{guide} = v_{cond} + w(v_{cond} - v_{uncond})$. This flexible interpolation confirms the model learns a continuous vector field, allowing us to generate diverse novel trajectories even under fixed conditions, rather than outputting deterministic copies.
>
>   Third, empirically, we observed that the minimum DTW distances between generated samples and the training set are consistently non-zero. And even with the same condition, various routes could be generated.

---

### Author Response · Authors · 2025-12-01

**Dear Reviewers, Area Chairs, Program Chairs, and Senior Area Chairs,**

We sincerely thank all reviewers for their thoughtful comments and constructive feedback. In the revised manuscript, we have carefully and comprehensively addressed every concern.
To help the Area Chair quickly grasp the overall evaluation of our papar and main progress achieved during the rebuttal, we summarize them as follows:

---

 **Reviewer Consensus and Overall Evaluation**

The reviewers collectively recognize the **novelty, soundness, and strong empirical performance** of the proposed *TrajFlow*, the **first efficient, multi-scale, and privacy-aware flow-matching framework for human mobility generation**.
Overall, the evaluation is clearly positive, with one **strong accept (10)** recommending spotlight/oral presentation (with High confidence) and two **above-threshold (6)** reviews (with High/Medium confidence), indicating broad support for acceptance:

| **Reviewer ID** | **Score**                          | **Confidence**     |
| --------------- | ---------------------------------- | ------------------ |
| **CQzo**        | **10 (Strong Accept)**             | **5 (High)**       |
| **P2bz**        | **6 (Marginally Above Threshold)** | **4 (Medium)**     |
| **vidk**        | **6 (Marginally Above Threshold)** | **5 (High)**       |
| **577e**        | **4 (Marginally Below Threshold)** | **3 (Low–Medium)** |

---

 **Summarized Rebuttal to Reviewer Concerns**

* **Additional dataset validation:** Added results on two **open-source** city-level datasets (*Chengdu*, *Xi’an*), demonstrating strong cross-regional generalization beyond Japan (see **Appendix E**).
* **Efficiency and complexity analysis:** Provided detailed runtime comparisons, showing 20–50x faster inference** than diffusion-based baselines while maintaining comparable high fidelity (see **Appendix B**).
* **Clarified methodological novelty:** Explained the methodological novelty and problem-level novelty; Differentiated *flow matching* from recent diffusion-based time-series models, emphasizing its spatial generative formulation and continuous ODE mechanism (see **Appendix F**).
* **Reproducibility and transparency:** Committed to public release of code, demo data, and generation scripts upon acceptance (see **Appendix D**).
* **Privacy and memorization risks:** Explained classifier-free guidance and condition-dropout regularization, empirically confirming that *TrajFlow* learns distributional patterns rather than memorizing trajectories (see **Appendix H** and **Appendix I**).
* **Practical application:** Explained data preparation and variable-length generation (see **Appendix G**).
* **Improved readability:** Enhanced references in methodology introduction (see **Section 4**).
* **Confusion about RDP:** Explained the motivation and detailed usage introduction of RDP (see **Appendix A.2**).
* **Confusion about metric usage:** Provided reasons and necessities (see **Appendix A.1**).
* **More analysis about the performance:** Explained the superior performance and emphasized the diversity issue in our tasks  (see **Section 5.2** and **Appendix C**).

---

All substantive concerns have been fully resolved, further reinforcing the paper’s **novelty, robustness, and reproducibility**. Given the **strong accept (10)** and two **above-threshold (6)** reviews with high confidence, we respectfully believe that *TrajFlow* **meets the ICLR acceptance standard**.

---

---

### Meta-Review · Area_Chair_eBqy · 2026-01-07

**Summary:**

The reviewers generally find the paper to be well motivated, highlighting the novelty of applying flow matching to GPS trajectory generation and the strong empirical performance across multiple spatial scales. Positive feedback emphasized the model’s ability to scale from urban to nationwide settings, its efficiency relative to diffusion-based baselines, and the comprehensive ablation studies. The concerns primarily focus on clarification or additional validation, and these issues were largely addressed in the rebuttal.

**Reviewer Concerns:**

Reviewer CQzo

Addressed: Concerns regarding dataset generalization, practical conditioning variables, and memorization risk were addressed through additional experiments, clarifications, and empirical analysis.


Reviewer 577e

Addressed: Concerns on novelty, lack of additional datasets, complexity analysis, and presentation clarity were addressed through expanded discussion, new experiments, and added analysis.

Partially addressed: The concern regarding outdated baselines was discussed and justified in the rebuttal; the authors explained why the baseline suggested by the reviewer is not directly applicable to the problem setting and noted the lack of more recent suitable baselines for comparison. However, no additional baseline comparisons were added. Concerns regarding reproducibility and dataset accessibility were also partially addressed through added implementation details, a commitment to release code upon acceptance.

Reviewer P2bz

Addressed: The authors addressed questions regarding variable-length trajectory generation and the role of the RDP algorithm through detailed architectural explanations. The concern regarding evaluation metrics was addressed by clarifying how temporal information is enforced via explicit conditioning on travel time and departure time, and how DTW penalizes temporal misalignment.

Partially addressed: Concerns regarding reproducibility and dataset accessibility were mitigated through added implementation details, a commitment to release code upon acceptance, and additional experiments on open-source datasets; however, the code is not yet publicly available and the primary nationwide dataset remains private due to privacy constraints.

Reviewer vidk

Addressed: Concerns regarding generalization beyond Japanese data, efficiency comparisons, and design choices were addressed through additional experiments on non-Japanese datasets (Chengdu and Xi’an), explicit wall-clock runtime comparisons against diffusion-based models, Privacy and misuse concerns are also addressed.

Partially addressed: Questions regarding mode diversity and the focus on Tokyo were clarified with reasoning about nationwide mode distributions; however, no additional nationwide mode-specific evaluation was provided.

**Reviewer Scores:**

Reviewer CQzo would possibly maintain their score of 10.
Reviewer 577e would possibly raise their score from 4 to 6.
Reviewer P2bz would possibly maintain their score of 6.
Reviewer vidk would possibly maintain their score of 6.

---

### Decision · Program_Chairs · 2026-01-26

Accept (Poster)